# An Integration Method for Regional PM_2.5_ Pollution Control Optimization Based on Meta-Analysis and Systematic Review

**DOI:** 10.3390/ijerph19010344

**Published:** 2021-12-29

**Authors:** Bingkui Qiu, Min Zhou, Yang Qiu, Yuxiang Ma, Chaonan Ma, Jiating Tu, Siqi Li

**Affiliations:** 1Department of Tourism Management, Jin Zhong University, Jinzhong 033619, China; qbk@jzxy.edu.cn; 2College of Public Administration, Huazhong University of Science and Technology, Wuhan 430074, China; myx0996@163.com (Y.M.); 17863935943@163.com (C.M.); tujiating1999@163.com (J.T.); withdoca@163.com (S.L.); 3Department of Economics, University of Warwick, Coventry CV4 7AL, UK; yang.qiu.21@ucl.ac.uk

**Keywords:** meta-analysis, two-objective optimization, air quality management, mortality rate, Nanshan district

## Abstract

PM_2.5_ pollution in China is becoming increasingly severe, threatening public health. The major goal of this study is to evaluate the mortality rate attributed to PM_2.5_ pollution and design pollution mitigation schemes in a southern district of China through a two-objective optimization model. The mortality rate is estimated by health effect evaluation model. Subjected to limited data information, it is assumed that the meta-analysis method, through summarizing and combining the research results on the same subject, was suitable to estimate the percentage of deaths caused by PM_2.5_ pollution. The critical parameters, such as the total number of deaths and the background concentration of PM_2.5_, were obtained through on-site survey, data collection, literature search, policy analysis, and expert consultation. The equations for estimating the number of deaths caused by PM_2.5_ pollution were established by incorporating the relationship coefficient of exposure to reaction, calculated residual PM_2.5_ concentration of affected region, and statistical total base number of deaths into a general framework. To balance the cost from air quality improvement and human health risks, a two-objective optimization model was developed. The first objective is to minimize the mortality rate attributable to PM_2.5_ pollution, and the second objective is to minimize the total system cost over three periods. The optimization results demonstrated that the combination of weights assigned to the two objectives significantly influenced the model output. For example, a high weight value assigned to minimizing the number of deaths results in the increased use of treatment techniques with higher efficiencies and a dramatic decrease in pollutant concentrations. In contrast, a model weighted more toward minimizing economic loss may lead to an increase in the death toll due to exposure to higher air pollution levels. The effective application of this model in the Nanshan District of Shenzhen City, China, is expected to serve as a basis for similar work in other parts of the world in the future.

## 1. Introduction

With the rapid industrialization and urbanization and continuous improvement of human infrastructure, many pollutants (such as PM, SO_2_, and CO) are released into the atmosphere, deteriorating the living environment. The severity of air pollution has become a global issue, playing a significant role in economic and environmental policymaking. Among many atmospheric pollutants, PM_2.5_ is believed to be the main culprit behind human morbidity/mortality [1]. Due to its small particle size, long transmission range, long half-life, and overall toxicity, it can damage the human circulatory system [2,3]. The Global Burden of Disease Report indicates that the number of deaths caused by long-term exposure to PM_2.5_ in 2019 was about 4.2 million worldwide, accounting for 7.6 percent of the total deaths that year. The severity of air pollution, especially the health threat of PM_2.5_, was emphasized in this paragraph.

Recently, the magnitude of PM_2.5_ pollution in China is becoming exceedingly serious. According to the aerosol inversion data from NASA satellites, the annual average concentration of PM_2.5_ in China reached 80 μg/m^3^ between 2006 and 2015, eight times the recommended air quality standard of the World Health Organization (WHO). PM_2.5_ pollution in China is characterized by a high annual and daily average concentration, resulting in serious social and economic losses. The resulting Disability-Adjusted Life-Years (DALYs) loss in China amounted to USD 16.09 million or 4.2 percent of the global DALYs loss. The increase in PM_2.5_ concentration increases the risk of disease and death from arteriosclerosis, heart disease, cerebral infarction, lung cancer, asthma, and chronic bronchitis. PM_2.5_ has become the fifth major factor of mortality in China [4]. Therefore, it is critical to evaluate the mortality effects of PM_2.5_ pollution to promote an air pollution prevention strategy that is beneficial to people’s health in China. The pollution degree of PM_2.5_ and its health damage in China were identified, which reflected the importance of health effect evaluation of this pollutant.

At present, many large and medium-sized cities in China have carried out urban air quality monitoring and forecasting. A series of measures have been taken to effectively control air pollution, including relocating factories away from residential areas, gradually prohibiting the utilization of small coal-fired boilers, and promoting cleaner energy. Additional steps such as improving the emission standards of automobile exhaust gas aimed at fundamentally controlling and/or eliminating severe air pollution in urban areas were also taken. Long-term control measures are restricted by their own characteristics, including long implementation cycles and slow effects. Therefore, it is necessary to flexibly control air pollution based on short-term effective treatment techniques to complement the long-term control measures. It clarified the importance of short-term treatment techniques and laid the foundation for the subsequent application of optimization model.

## 2. Literature Review

### 2.1. Summary of the Health Evaluation Attributed to PM_2.5_ Pollution

Several existing studies have investigated the health impact of exposure to PM_2.5_ in the fields of toxicology, epidemiology, environmental science, economics, and geography. In epidemiology, initial studies mainly evaluated the health effects of PM_2.5_ pollution in their sample populations, which was then used as the basis for assessing the health effects of PM_2.5_ in general [5]. The selected health endpoints in such studies included total mortality, respiratory disease mortality, cardiovascular disease mortality, respiratory disease incidence, and cardiovascular disease incidence. It has been confirmed that exposure to increased PM_2.5_ concentrations leads to an increase in mortality; however, due to differences in regional selections, research methods, and sample sizes between various studies, the exposure–response coefficients of different epidemiological cases exhibited a huge discrepancy [6,7,8,9,10]. The existing research results show that the meta-analysis method can systematically evaluate and quantitatively analyze the health hazards caused by PM_2.5_ pollution by examining the existing results [8]. A large number of epidemiological studies makes it possible to use meta-analysis to accurately calculate the exposure response coefficient, laying a foundation for quantifying the health effects of PM_2.5_ pollution [9,11,12,13,14,15,16,17,18,19,20,21,22,23,24,25,26,27,28,29,30,31,32,33,34,35,36,37]. For example, Fu et al. [19] found a significant association between PM_2.5_ exposure and stroke, dementia, Alzheimer’s disease, and Parkinson’s disease. Luo et al. [21] evaluated the effect estimates of the relationship between short-term exposure to PM_10_ and PM_2.5_, and risk of myocardial infarction through the meta-analysis approach. Liu et al. [27] calculated the long-term effect of exposure to ambient PM on overall CVD (i.e., cardiovascular disease) mortality according to the WHO’s interim targets. The study demonstrated that the long-term ambient PM_2.5_ exposure level was positively associated with overall CVD mortality. Therefore, governments should exert greater effort to improve air quality given its adverse health implications. The studies mentioned above provide a reference for research on the health effects of PM_2.5_ pollution and a basis for policymakers to formulate evidence-backed policies for dealing with the issue. However, these studies mainly focus on the present situation rather than mitigating air pollution and rarely address the design and implementation of pollution control measures. The successful application of the meta-analysis method in health effect evaluation demonstrated its feasibility and reliability. The health threat caused by PM_2.5_ pollution should be incorporated into the pollution control management.

### 2.2. Summary of Pollution-Mitigation Optimization Aiming at Air Pollutants 

Environmental engineering research uses optimization techniques to address the mitigation of PM_2.5_ pollution. This research often focuses on designing a suitable combination of pollution control measures by establishing the quantitative relationship between the emission of harmful substances and the atmosphere and cost and benefit [38,39,40,41,42,43,44,45,46,47]. For example, Carnevale et al. [38] established a nonlinear optimization model and applied it to the study of O_3_ pollution control in large cities in northern Italy; the model focused on the decontamination costs and improvement of the air quality index. Pisoni and Volta [40] formulated a two-objective optimization model for tackling the pollution control issue of PM_10_, where the health impacts of PM_10_ pollution were considered as external cost. Zhen et al. [41] developed an interval-parameter fuzzy programming mixed integer programming method for supporting the energy systems management and air pollution mitigation control under multiple uncertainties. Huang et al. [47] developed a multi-pollutant cost–benefit optimization system based on a genetic algorithm for generating regional air quality control strategies. Although such studies have made significant advancements in regional air pollution control, management task allocation, and cooperative income distribution, they still fall short of the goal of minimizing population health damage. In sum, there are advantages of optimization models for pollution control and its potential improvement.

Therefore, based on a comprehensive view of existing epidemiological and environmental engineering studies, the lacunae in the research work consists of the following two parts: (i) the use of meta-analysis to analyze the results of multiple studies on PM_2.5_ pollution and the population health in recent years to systematically and quantitatively define the relationship coefficient of PM_2.5_ pollutant exposure to population death and accomplish the health impact assessment; (ii) establishing a two-objective optimization model to determine the most reasonable and effective treatment plans, taking into account constraints such as socio-economic activity, environmental quality, technical feasibility, and cost. This study focused on the Nanshan District of Shenzhen in southeast China, which is expected to serve as a basis for similar work in other parts of the world in the future. Two important parts of this research work were described in this paragraph.

## 3. Method

In this section, Section 3.1 introduces the formulation process of the health evaluation model based on the meta-analysis method. Four critical steps, including the literature search, literature screening, determination of exposure-response relationship coefficient, and the calculation of mortality caused by PM_2.5_ pollution, are described in detail. These are expected to set a good example for the application of meta-analysis method in other cases. Section 3.2 describes the formulation and solution process of the two-objective optimization model. The detailed introduction of model components, including two objective functions and all constraints, are provided in this section. Moreover, the conversion of two-objective optimization model to single-objective one and the LINGO software (Shenzhen, China) are also introduced to facilitate applications in other regions.

### 3.1. The Establishment of Health Evaluation Model Using Meta-Analysis Method

With the presence of the PM_2.5_ health effect database in China, it is necessary to collect new data from previous research and use meta-analysis to derive a robust estimate of the exposure-response coefficient of atmosphere PM_2.5_ to population mortality and, finally, construct the PM_2.5_ health effect evaluation model. The health hazards caused by atmospheric PM_2.5_ pollution were quantitatively evaluated, providing a scientific basis for relevant policymaking. The meta-analysis method is firstly used to summarize and combine research results on the same subject under specific conditions. The main function of the meta-analysis method was introduced. The main procedure and steps of the meta-analysis method are presented as follows.

#### 3.1.1. Literature Search

Endnote literature retrieval was used as a basis for meta-analysis. The following terms were searched within the title, abstract, or keywords of both English and Chinese studies: ‘PM_2.5_’, ‘particulate matter’, ‘ambient air pollution’, ‘mortality’, ‘dose–response’, and ‘short-term health effect’. We searched several academic databases such as the Chinese Periodical Network (CNKI), PubMed, and Web of Science to ensure that only the peer-reviewed scientific literature was selected. From these sources, we collected the epidemiological literature on the relationship between PM_2.5_ pollution and daily mortality of residents published worldwide between 2000 and 2019 and extracted the exposure-response relationship coefficient at the 95% CI (i.e., confidence interval). It is noticed that the statistical analysis model involved in the selected literature was based on the time series of Generalized Additive Model (GAM) and Generalized Linear Model (GLM). 

#### 3.1.2. The Inclusion and Exclusion Criteria of Candidate Literature Studies

A comprehensive literature search was conducted to obtain an estimate based on recent epidemiological studies worldwide according to the following criteria: (i) the endpoint of the health effect involved in the literature was defined as the total mortality of residents studied, and the exposure–response relationship was established as being between the total mortality of the residents and the PM_2.5_ concentration; (ii) upon review, studies with redundant information, those that were based on data from earlier studies published before 2000, or those without data on relative error and conclusions drawn with at least a 95% CI were excluded; (iii) all eligible data were statistically entered into our database of the relationship between PM_2.5_ and human health effects, as an exposure-response relationship, where it was expressed as the percentage of the change of total mortality when the concentration of PM_2.5_ increased by 10 μg/m^3^. According to the inclusion and exclusion criteria mentioned above, we found and assessed 15 articles (five in Chinese and ten in English), including 20 datasets, covering 12 research areas in China, shown in Table 1 [48,49,50,51,52,53,54,55,56,57,58,59,60,61,62]. The process of literature screening is described in this paragraph.

#### 3.1.3. The Heterogeneity Analysis and the Determination of Exposure-Response Relationship Coefficient

This analysis utilized the ‘metagen’ command of the ‘meta package’ in ‘R’ software. When extracting the data from the literature, the resident mortality rate that varied with an increasing PM_2.5_ concentration and its relative standard error (SX) was directly used to test the consistency of the database results. If the homogeneity test was accepted, the fixed-effect model was selected. Conversely, if the homogeneity test was rejected, the results were inconsistent, and a stochastic utility model was used. The result was described as the percentage increase in the total death rate per 10 μg/m^3^ increase in PM_2.5_, expressed as *β*. Since the p-value was less than 0.0001, and I^2^ = 100%, the random-effects model was selected. The difference was statistically significant, indicating heterogeneity among the studies. Using the random-effects model, we found lower-bound and upper-bound values of 0.0033 and 0.0067, respectively, indicating that the percentage of total mortality increased by 0.47% when the PM_2.5_ concentration increased by a certain unit (10 μg/m^3^ in this case). The software implementation of meta-analysis method and its output result are provided in this paragraph.

#### 3.1.4. The Calculation of Mortality Caused by PM_2.5_ Pollution

Equations (1) and (2) were used to calculate the mortality rate of residents from exposure to PM_2.5_ pollution. The major parameters involved in this process were sourced from several ways. For example, the relationship coefficient of exposure to reaction (i.e., *β*) was generated by meta-analysis method; the average PM_2.5_ concentration of affected areas was calculated based on precursors’ conversion ratios to PM_2.5_ and Gaussian dispersion model, where the related description of Gaussian dispersion model can be referred to Appendix A; total base number of deaths for affected areas was collected from local statistical yearbook.
(1)$$I=I_{0}\times \exp\left[\beta \times \left(C-C_{0}\right)\right]$$
(2)$$\Delta I=I-I_{0}=I\left\{1-1/\exp\left[\beta \times \left(C-C_{0}\right)\right]\right\}$$
where *β* is the relationship coefficient of exposure to reaction, which equals to 0.0047; *C* is the daily average PM_2.5_ concentration, which is calculated based on the conversion ratio between precursor pollutants (i.e., PM, SO_2_, and NO_x_) and PM_2.5_ and the Gaussian dispersion model. *C*_0_ is the reference concentration of PM_2.5_, *I* is the health effect under actual air pollution concentration, *I*_0_ is the population health effect under the reference concentration, and △*I* is the number of deaths caused by the excessive concentration of PM_2.5_. By referring to the Shenzhen Nanshan District statistical yearbook, Shenzhen Health Statistical Yearbook, and Shenzhen Statistical Yearbook, particularly those belonging to the year 2018, the latest resident mortality data (including the total number of deaths and average mortality rate) of Shenzhen residents were obtained. The total base number of deaths for the Nanshan District was estimated as 1353 people. In addition, the reference concentrations available for Shenzhen include the natural background concentrations of air pollutants, the minimum concentration observed in the past epidemiological literatures, and the hygienic standards established by government agencies. In this study, considering the present air quality in Nanshan District and the air quality requirement in the future, the reference concentration was determined to be 10 μg/m^3^. 

### 3.2. The Formulation of the Two-Objective Optimization Model

A multi-objective optimization model aiming at minimizing the total pollution control cost and the number of deaths caused by PM_2.5_ pollution was developed under the constraints of environmental quality, and the economic and technical feasibility of control measures. The model is formulated as follows:

#### Objective Function

(i) Minimization of the mortality rate attributable to PM_2.5_ pollution(3)$$Minimize\;f_{1}=\sum_{p=1}^{P}\sum_{k=1}^{K}\Delta I_{pk}$$(ii) Minimization of total system cost over three periods(4)$$Minimize\;f_{2}=\sum_{i=1}^{I}\sum_{j=1}^{J}\sum_{k=1}^{K}c_{jk}^{}x_{ijk}^{}+\sum_{i=1}^{I}\sum_{s=1}^{S}\sum_{k=1}^{K}c_{sk}^{}x_{isk}^{}+\sum_{i=1}^{I}\sum_{n=1}^{N}\sum_{k=1}^{K}c_{nk}^{}x_{ink}^{}$$ where *f*_1_ is total number of deaths, *p* is the affected area, *P* is total number of affected area, *f*_2_ is total system cost, *i* is the type of pollution source, *I* is total number, *j* is the type of control measure for PM (total suspended particulate), *J* is total number, *k* is planning period, *K* is total number, *c_jk_* is the operational cost of the control measure *j* during the period *k*, *x_ijk_* is total PM amount allocated from the emission source *i* to the control measure *j* during the period *k*m *s* is the control measure for sulfur dioxide (i.e., SO_2_), *S* is total number, *c_sk_* is the operational cost of the control technique *s* during the period *k*, *x_isk_* is treated SO_2_ amount sourced from the emission source *i* of the control technique *s* during the period *k*, *n* is the control measure for nitrogen dioxide (i.e., NO_x_), *N* is total number, *c_nk_* is the operational cost of the control measure *n* during the period *k*, *x_ink_* is total NO_x_ amount sourced from the emission source *i* disposed by the control measure *n* during the period *k*. 

This function is subject to:(I) The limitations in the pollutant treatment
(5)$$\sum_{j=1}^{J}x_{ijk}^{}=G_{idk}^{}{}_{,\;\;}\forall i,k$$
(6)$$\sum_{s=1}^{S}x_{isk}^{}=G_{iok}^{}{}_{,\;\;}\forall i,k$$
(7)$$\sum_{n=1}^{N}x_{ink}^{}=G_{ink}^{}{}_{,\;\;}\forall i,k$$
where *G_idk_* is PM emitted by source *i* during the period *k*, *G_iok_* is SO_2_ emitted by source *i* during the period *k*, and *G_ink_* is NO_x_ emitted by source *i* during the period *k*.
(II) The regulations of the emission sources
(8)$$\sum_{j=1}^{J}(1-\eta_{j})x_{ijk}\leq e_{idk}\;\forall i,k$$
(9)$$\sum_{s=1}^{S}(1-\eta_{s})x_{isk}\leq e_{isk}\;\forall i,k$$
(10)$$\sum_{n=1}^{N}(1-\eta_{n})x_{ink}\leq e_{ink}\;\forall i,k$$where *η_j_* is the removal efficiency of PM for the control measure *j*. *e_idk_* is the permissible PM emission for the pollution source *i* during the period *k*; *η_s_* is the removal efficiency of SO_2_ for the control technique *s*. *e_isk_* is the permissible SO_2_ emission for the pollution source *i* during the period *k*; *η_n_* is the removal efficiency of NO_x_ for the control measure *n*. *e_ink_* is the permissible NO_x_ emission for the pollution source *i* during the period *k*.

(III) The constraints of environmental load capacity:(11)$$\sum_{i=1}^{I}\sum_{j=1}^{J}t_{ip}\rho_{d}(1-\eta_{j})x_{ijk}+\sum_{i=1}^{I}\sum_{s=1}^{S}t_{ip}\rho_{s}(1-\eta_{s})x_{isk}+\sum_{i=1}^{I}\sum_{n=1}^{N}t_{ip}\rho_{n}(1-\eta_{n})x_{ink}\leq a_{pk},\;\;\forall p,k$$where *t_ip_* is the transfer coefficient from the emission source *i* to the receptor zone *p*, which is evaluated using Gaussian dispersion model and Pasquill–Gifford curves; *ρ_d_* is the conversion ratio between PM and PM_2.5_; *ρ_s_* is the conversion ratio between SO_2_ and PM_2.5_; *ρ_n_* is the conversion ratio between NO_x_ and PM_2.5_. The above three conversion ratios were determined based on chemical composition analysis and source apportionment results provided by local environmental protection agency and study result of relevant research [63,64,65,66]. *a_pk_* is the maximum allowable PM_2.5_ concentration of the receptor zone *p* during the period *k*.
(IV) Nonnegative constraints:
(12)$$x_{ijk},x_{isk},x_{ink}\geq 0,\;\forall i,j,s,n,k$$

Next, the weight summation approach was used to address the two objectives (3) and (4). New objective function (13) was established by allocating two weight coefficients (*w*_1_ and *w*_2_) to the original two objectives (*f*_1_ and *f*_2_), where the comparison between two coefficients reflected the relative importance of the two objectives.
(13)$$Minimize\;f=w_{1}\frac{f_{1}-f_{1,\min}}{f_{1,\max}-f_{1,\min}}+w_{2}\frac{f_{2}-f_{2,\min}}{f_{2,\max}-f_{2,\min}}$$
where *f*_1,min_, *f*_1,max_, *f*_2,min_, and *f*_2,max_ are the minimum and maximum possible values obtained through solving the optimization model, while two objectives *f*_1_ and *f*_2_ are solved as a single objective function, respectively.

Subject to:(14)$$\sum_{j=1}^{J}x_{ijk}^{}=G_{idk}^{}{}_{,\;\;}\forall i,k$$
(15)$$\sum_{s=1}^{S}x_{isk}^{}=G_{iok}^{}{}_{,\;\;}\forall i,k$$
(16)$$\sum_{n=1}^{N}x_{ink}^{}=G_{ink}^{}{}_{,\;\;}\forall i,k$$
(17)$$\sum_{j=1}^{J}(1-\eta_{j})x_{ijk}\leq e_{idk}\;\forall i,k$$
(18)$$\sum_{s=1}^{S}(1-\eta_{s})x_{isk}\leq e_{isk}\;\forall i,k$$
(19)$$\sum_{n=1}^{N}(1-\eta_{n})x_{ink}\leq e_{ink}\;\forall i,k$$
(20)$$\sum_{i=1}^{I}\sum_{j=1}^{J}t_{ip}\rho_{d}(1-\eta_{j})x_{ijk}+\sum_{i=1}^{I}\sum_{s=1}^{S}t_{ip}\rho_{s}(1-\eta_{s})x_{isk}+\sum_{i=1}^{I}\sum_{n=1}^{N}t_{ip}\rho_{n}(1-\eta_{n})x_{ink}\leq a_{pk},\;\;\forall p,k$$

Finally, the optimal treated amounts of three types of pollutants corresponding to various control measures were provided by solving the single-objective model. The overall framework of the model is shown in Figure 1. The model is a typical linear programming problem, which can be solved using various tools and software, such as Excel, Lingo, Matlab, and the LINGO software. This study used the LINGO software (version 11.0) developed by LINDO Systems Incorporation. It has many advantages, including convenient input, user-friendly operation, and fast running speed when solving linear programming, integer programming, and other programming problems. LINGO has been widely used in various scientific modeling fields for many years. Section 3.2 described the formulation and solution process of two-objective optimization model. The definition of two objective functions, the implications of all constraints and the explanations of model parameters are provided in this section. The conversion of the two-objective optimization model to the single-objective one and the LINGO software are also introduced in this section.

## 4. Case Study

### 4.1. Introduction of the Study Area 

The Nanshan District is located in the western part of the Shenzhen Special Economic Zone. Figure 2 demonstrates the location of Nanshan District. This district faces Shenzhen Bay in the east, Pearl River Estuary in the west, Dachan Island in the south, and Yuen Long, Hong Kong, across the sea. Nanshan district has a total area of 192 km^2^, a total population of 1.5 million, and gross domestic product of USD 78.13 billion in 2019. Thus, it is evident that this district is a densely populated region with large-scale economic development. In recent years, along with rapid urbanization and industrialization in the Nanshan District, various environmental problems have also emerged. The annual average values of SO_2_ and PM_10_ in the Nanshan District have reached 0.01 and 0.05 mg/m^3^, respectively, with yearly growth rates of approximately 15% and 7%, respectively [67]. This phenomenon is largely attributed to the production and supply of electricity and heating industries, which accounted for 91.95% of the total emissions. Notably, the Mawan power plant and the Nanshan thermal power plant in this district have been identified as the largest contributors to exhaust gas pollutants (i.e., SO_2_ and dust pollutants), accounting for 70% of total atmospheric pollutant emissions in the entire city. Therefore, mitigating atmospheric pollutant emissions in the Nanshan District could improve the air quality not only in the district but also throughout Shenzhen city. The partial pollutant treatment technologies and measures are unable to tackle regional air pollution effectively. Thus, developing system engineering and analysis technologies have promising applications in the field of environmental pollution control. 

### 4.2. The Utilization of System Engineering Technology 

System analysis is the act of performing qualitative and quantitative research on a regional system or facility system. Figure 3 reflects the specific components and processes of air quality management in the Nanshan district based on the system engineering technology. 

#### 4.2.1. The Investigation and Analysis of the System Status

Firstly, the study area’s social, economic, and environmental situation was estimated to define the direction for future development and identify the major issues that should be focused on during the regional development process. The investigation results indicated that the Nanshan District should primarily focus on improving economic growth, expanding the production scale of its enterprises, and striving for further development of its society and economy in concert with technological innovations. However, previous experience shows that when regional economic development is promoted by production expansion, a series of environmental and public health issues also emerge. Recently, the frequent haze phenomenon has attracted the attention of the local authorities. Therefore, it is essential to determine the enterprises’ development scale and pollutant-reduction amounts to facilitate coordinated development of the society, economy, and the environment by minimizing economic costs and the population death toll. 

#### 4.2.2. The Determination of System Boundary 

The system boundary mainly comprises determining the scope of time and space, which is the key to formulate the optimization model for tackling practical problems. This is because the number and types of parameters are directly dependent on the determination of the space scope. The overlarge scope may result in many parameters and complex relationships, which is adverse to formulate and solve the optimization model. Conversely, oversimplification leads to the optimization model being inconsistent with the actual system situation, which may influence the effectiveness. Therefore, based on the site survey and available data, the spatial scope for our system was defined as the entire Nanshan District (22°24′~22°39′ north latitude and 113°47′~114°01′ east longitude). The time scope was set to three years (2020–2022), which was further split into three planning periods with each period including one year (i.e., k = 1 for 2020, k = 2 for 2021 and k = 3 for 2022).

#### 4.2.3. The Identification and Analysis of System Elements

Ideally, all PM_2.5_ sources (including primary, secondary, fixed, mobile, local, and external sources) in the region should be considered as the control target. However, this may cause excessive computational burden and lead to unfavorable control schemes for external and mobile sources. Therefore, five major fixed emission sources, including Shenzhen Nanshan Thermal Power corporation, Shenzhen Mawan Power corporation, Nanshun Oil corporation, Guangdong Yaopi Glass corporation, and Shenzhen Huajing Glass corporation, were chosen as the major control sources. The on-site survey and statistical results provided by EIA (Environmental Impact Assessment) reports demonstrated that the pollutants discharged from the above-mentioned five sources mainly included PM, SO_2_, and NO_x_, which are considered the precursor pollutants that contributed to PM_2.5_. It was assumed that the remaining sources met the emission standards. The contribution of such sources to ambient PM_2.5_ was estimated based on the percentage composition of the source identification. According to the local scenario and operational cost and efficiency of related technologies, four types of control technologies were selected for three types of pollutants that contributed to PM_2.5_. Among them, the dust removal technology includes the bag collector (BF), cyclone collector (CL), wet collector (WS), and electrostatic collector (EP). The desulfurization technologies were composed of limestone-gypsum (LG), spray drying (SD), circulating fluidized bed (CFB), and limestone injection (LI), respectively. The denitration technologies consisted of selective catalytic reduction (SCR), selective non-catalytic reduction (SNCR), SCR + SNCR, and low nitrogen combustion (LNB) + SNCR technologies. In addition, according to the distribution of functional zones, it was determined that the affected areas included four categories, namely residential areas, industrial areas, health resorts, and scenic areas. The relationship between the emission intensity of pollutant sources and pollutant concentration within the affected region was estimated using a Gaussian diffusion model.

#### 4.2.4. The Critical System Parameters

The overall objective of this study was to facilitate collaborative development amongst social, economic, and environmental factors using an optimization model. The same was done using predetermined system boundaries, combined with the development goal and environmental problems of the Nanshan district. Table 2 provides the relevant information of five sources, which are mostly obtained from on-site investigation and data statistics. Moreover, their discharge standards were regulated by an environmental protection agency based on the types of industries. Each emission source should ideally be equipped with effective emission reduction measures to satisfy the ever-increasing demand for improved environmental quality. The treatment cost and efficiency of candidate technologies (as shown in Table 3) were mainly determined based on the field investigation, literature review, and expert consultation. Given the impact of discount rates and other factors, the costs are expected to rise in the future. Moreover, as the living standard continues to improve, the pollution load capacity of the affected region should be maintained at a lower level and decreased gradually. In this study, the annual mean concentration of PM_2.5_ regulated by the Ambient Air Quality Standards released by China (GB3095-2012) was used as a reference. The residential and industrial regions execute the secondary standards with concentration limits of 35, 31.5, and 28 μg/m^3^ for the three periods. In contrast, the other two regions are compelled to use the primary standards of 15, 13.5, and 12 μg/m^3^ during the three periods. From the perspective of system analysis, Section 4.2 provides the detail information on system status, boundary, elements, and critical parameters.

## 5. Results and Discussion

### 5.1. Result Analysis

Table 4, Table 5 and Table 6 provide the model solutions under different weight combinations, presented as the treatment quantity for various candidate control technologies over three planning periods. Based on the weight summation approach, the priority is to set diverse weight combinations, i.e., assigning different values to *w*_1_ and *w*_2_, where the former is used to reflect the importance of population health, while the latter indicates the importance of minimizing economic costs. Generally, a high weight value denotes the higher significance of the relevant objective, with *w*_1_ plus *w*_2_ always equal to 1. A total of nine weight combinations were evaluated in this study, which were *w*_1_ = 0.1 and *w*_2_ = 0.9, *w*_1_ = 0.2 and *w*_2_ = 0.8, *w*_1_ = 0.3 and *w*_2_ = 0.7, *w*_1_ = 0.4 and *w*_2_ = 0.6, *w*_1_ = 0.5 and *w*_2_ = 0.5, *w*_1_ = 0.6 and *w*_2_ = 0.4, *w*_1_ = 0.7 and *w*_2_ = 0.3, *w*_1_ = 0.8 and *w*_2_ = 0.2, and *w*_1_ = 0.9 and *w*_2_ = 0.1, respectively. The purpose of assessing different weight combinations was to reflect the impact of various combinations on the decision variables and objective values of model results. Among them, the decision variables under four scenarios were included in Table 4, Table 5 and Table 6. As illustrated in the three tables, different weights greatly influence the results, which is reflected in the quantity of treatment recommended for each candidate technology during different planning periods. This paragraph introduced the nine weight combinations and emphasized the influence of weight combination on the treatment scheme.

For example, when *w*_1_ = 0.7 and *w*_2_ = 0.3, the pollutant PMs generated from PP source over three periods are largely handled by BF with the highest efficiency and treatment cost, with treatment quantities of 3.84, 4.22, and 4.42 t/d, respectively. However, the weight combination of *w*_1_ = 0.3 and *w*_2_ = 0.7, WS, which has a lower treatment efficiency and treatment cost, plays a major role in dealing with the same pollutants, with treatment amounts of 3.84, 4.22, and 4.42 t/d, respectively. A similar situation also occurs for the Gc2 source, where the pollutants at *w*_1_ = 0.7 and *w*_2_ = 0.3 are disposed through BF, being 0.09, 0.09, and 0.10 t/d, respectively. On the contrary, as *w*_1_ is decreased (i.e., 0.3) and *w*_2_ is increased (i.e., 0.7), BF is unused. Correspondingly, the pollutant is treated by the WS technique. This difference is due mainly to the first weight combination implying that population health is the priority objective. Therefore, applying the BF technique, which has the highest efficiency and cost, is most preferred. Compared to the first weight combination, the second is more focused on the total system cost, leading to the WS technique playing a more central role. It reflected the influence of weight combination on the utilization of PM-treated technology.

In addition, this trend is also remarkably evident in selecting the treatment technology for the other two pollutants. Concerning SO_2_, an increased *w*_1_ value requires the technology with a higher treatment efficiency to be used more frequently (i.e., LG). When *w*_1_ = 0.7 and *w*_2_ = 0.3, the pollutant SO_2_ sourced from PPc source in three periods is treated by the LG technique, being 69.31, 76.24, and 79.71 t/d, respectively. Conversely, the lower-efficiency technology (CFB) plays the complementary role under *w*_1_ = 0.3 and *w*_2_ = 0.7 for minimizing the total cost, where, in the second period, the treatment amounts of the two techniques (i.e., LG and CFB) were 42.17 and 34.07 t/d, respectively; while those in the third period were 68.69 and 11.02 t/d, respectively. Similarly, the pollutant NO_x_ generated by PP source under *w*_1_ = 0.7 and *w*_2_ = 0.3 was treated using combined technology (i.e., LNB + SCR), with the highest treatment efficiency of 70.69, 77.75, and 81.29 t/d, respectively. Conversely, the SCR technique with the low treatment efficiency is a major option at *w*_1_ = 0.3 and *w*_2_ = 0.7. The same variation also appears in the Gc1 source. When *w*_1_ = 0.7 and *w*_2_ = 0.3, it is disposed of through the LNB + SCR technique over three periods, with the treated amounts of 0.67, 0.74, and 0.77 t/d, respectively. When *w*_1_ = 0.3 and *w*_2_ = 0.7, the amounts allocated to the combined technique in the first period is 0.67 t/d; those treated by SCR in the other two periods are 0.74 and 0.77 t/d, respectively. This reflects the influence of weight combination on the utilization of candidate treated technology of two pollutants (SO_2_ and NO_x_).

The tendency that emerged in total treated quantities of various techniques better reflects the influences of predetermined weight combinations on the treatment schemes than the results in a single period. As demonstrated in Table 4, it is obvious that along with *w*_1_ declining and *w*_2_ growing, the quantity of pollutants disposed of would decrease and increase accordingly for high-efficiency and low-efficiency technologies. Taking the PP source as an example, the total amount disposed of by BF over three periods under four weight combinations was 12.48, 12.48, 0, and 0 t/d, respectively. The quantities disposed of by WS were found to be 0, 0, 12.48, and 12.48 t/d, respectively. Similarly, for the Gc2 source, the total amount disposed of by BF generally showed a decreasing trend of 0.28, 0.28, 0, and 0 t/d, respectively; as opposed to the figures for WS, which showed an increasing trend of 0, 0, 0.28, and 0.28 t/d, respectively. This disparity can be largely attributed to the fact that the treatment cost and efficiency of BF are higher than those of WS. This describes the variation in total treated quantities of various techniques in order to better reflect the influence of predetermined weight combinations on the treatment schemes.

Figure 4 reflects the residual PM_2.5_ concentration after the treatment process of four affected regions over three periods under nine weight combinations. It is evident that the PM_2.5_ concentration in each period satisfies the respective air quality requirements based on the optimization model. With the decrease on *w*_1_ and increase on *w*_2_, the PM_2.5_ concentration in four regions showed a stringent increasing trend. For example, in the second period, the pollutant concentrations under all scenarios for residential and scenic regions were 13.43, 13.45, 13.45, 13.54, 13.54, 13.55, 13.79, 13.79, and 13.79 t/d, respectively, and 5.89, 5.93, 5.93, 6.16, 6.16, 6.19, 6.30, 6.30, and 6.30 μg/m^3^, respectively. A similar situation also occurred in the third period of the industrial area, with pollutant concentrations increasing from 13.35 to 15.39 μg/m^3^. The reason behind the increase in the PM_2.5_ concentration is that the decrease in *w*_1_ and increase in *w*_2_ results in more attention to the system economy than the population health. Therefore, the technologies with both inexpensive and low-efficiency abatement technologies were used more frequently. It can be concluded that the reduction of economic cost is at the expense of the increase in pollutant concentration, which in turn may adversely impact people’s health. For example, under nine different weighting scenarios, the mortality number in industrial areas is 19, 21, 22, 26, 26, 27, 35, 37, and 37 μg/m^3^, respectively. Similar growth in mortality is also reflected in the scenic area, with numbers increasing from 7 to 11 μg/m^3^ for *w*_1_. The variation in residual PM_2.5_ concentration of affected regions under nine weight combinations and the related reasons are analyzed in this paragraph.

In addition to decision variables, the influence caused by various weight combinations also affects the objective values. The variations in the death toll and total system cost under different weight combinations are depicted in Figure 5. It is evident that with the increase in *w*_2,_ the death toll exhibited a gradual increase from 80 to 122 Person. Conversely, total system cost decreased from 37 to 26.58 × 10^4^ USD, respectively. The above variations reflect a trade-off between system economy and health risk. The low treatment cost is accompanied by a high mortality rate. Conversely, the huge investment is favorable to reducing the pollutant concentration, thereby alleviating the damage to population health. Currently, population health issues have gained more attention with rapid economic growth. Therefore, the treatment scheme with high *w*_1_ and low *w*_2_ (*w*_1_ = 0.9 and *w*_2_ = 0.1) is recommended for generating the pollution control strategies. This paragraph described the variation in two objective functions and analyzed the trade-off between system economy and health risk.

### 5.2. Discussion

There are insignificant differences in the coefficient of the exposure–response relationship calculated using the meta-analysis approach in this study compared to other studies performed in areas adjacent to the study region. For example, the coefficient value for the Guangdong Province is reported to be 0.0060 [54,55,56,57,59,60,63]. The factors causing this difference mainly include the exposure level, chemical constituents of pollutants, urban sources, and population characteristics. For example, the higher tolerance to pollution exposure of people living in places with severe air pollution might reduce mortality. However, mortality would still increase significantly in the region as the population ages. Another possible factor causing the differences in the health evaluation results could come from the shape of the exposure–response function applied in this study. The function used in this study was derived from research conducted in the cities of developed countries with a relatively lower PM_2.5_ concentration. Other functions, such as the integrated exposure–response (IER) model [68,69] can also be incorporated into the optimization model. Finally, some parameters included in this optimization model had uncertainties associated with them, such as the pollutant emissions from sources, the treatment efficiency and operational cost of candidate control technique, and the transfer coefficient. These uncertainties were simplified as deterministic parameters in this study, possibly resulting in differences from other studies. Therefore, introducing various types of uncertainty optimization techniques—including interval, fuzzy, and stochastic programming models—may ensure the robustness and reliability of this optimization model. The potential improvement of the proposed health effect evaluation model and optimization model for pollution control is discussed in this paragraph.

## 6. Recommendation

The serious economic loss, health threat, and air-quality degradation caused by PM_2.5_ pollution has aroused a wide attention globally. The research results of this study revealed the necessity of combining the health effect evaluation based on meta-analysis method and the air pollution control with aid of the optimization model, and it emphasized the importance of the balance between the high economic cost and people’s health threat. Some recommendations were provided based on the research findings: (i) for the local residents, it is necessary to obtain an in-depth insight into the health damage caused by PM_2.5_ pollution, which is beneficial to strengthen awareness of human health and environmental protection; (ii) for the administrators of government agency, this study helps them to identify the major pollutants sources and design effective pollution-mitigation measures. It is suggested that the health risk evaluation be incorporated into the process of air pollution control planning and management, with emission and ambient air-quality standards being properly designed in order to avoid unnecessary socio-economic losses. 

## 7. Conclusions

Several measures, including keeping factories away from residential areas, gradually banning the use of small coal-fired boilers (or remolding them to use clean energy), as well as improving emission standards for automobile exhaust, have been taken by several cities in China for addressing severe air pollution in urban areas. Nevertheless, long-term control measures are flawed owing to their long implementation cycles and slow effects. Therefore, along with long-term control measures, there is an urgent need to devise short-term pollutants abatement methods. 

This study determines the best combination of treatment technologies through an optimization model for solving the air pollution issue. In contrast to the traditional pollution-control optimization models, this study used the meta-analysis method to estimate the increased percentage in the death toll caused by an increase in PM_2.5_ concentration. Additionally, the study also estimated the death toll in combination with preset background concentration, the total number of deaths, and actual contamination concentration. Death toll reduction, one of the major objectives of the optimization model, took the health loss into account while preserving the system’s economy. The model results indicate that the low health risk level could be achieved by employing high-efficiency treatment technologies, raising the reduction ratio of pollutants, and reducing the pollutant residues within the affected area, although this approach would cost a significant sum of money. People’s livelihood has gradually become the most valued issue in China, compared to economic development, as was previously the case. Therefore, the solution at *w*_1_ = 0.9 and *w*_2_ = 0.1 is recommended for helping to formulate PM_2.5_ reduction strategies. The effective application of this model in the Nanshan District of Shenzhen City, China, is expected to be a good example for similar work in other parts of the world in the future. The optimization model proposed in this study still needs to be improved in some aspects, such as selecting an appropriate exposure–response function and addressing uncertain optimization methods to tackle more complex issues in the future. 

## Figures and Tables

**Figure 1 ijerph-19-00344-f001:**
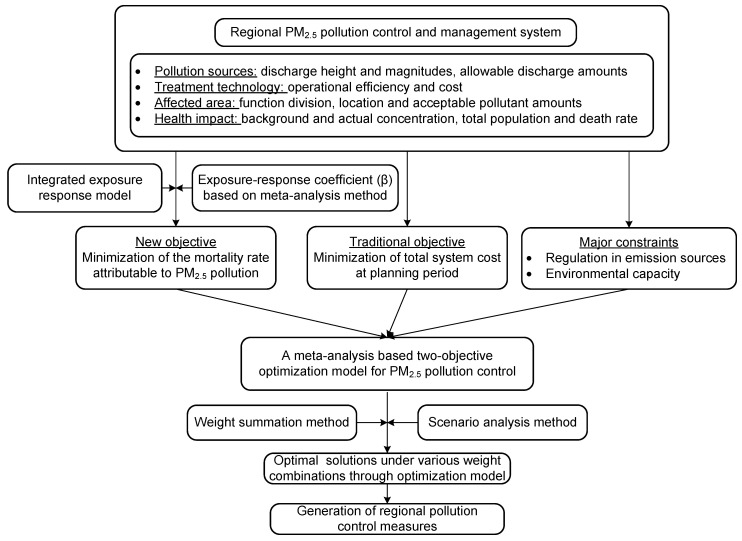
The formulation and solution framework of the proposed optimization model.

**Figure 2 ijerph-19-00344-f002:**
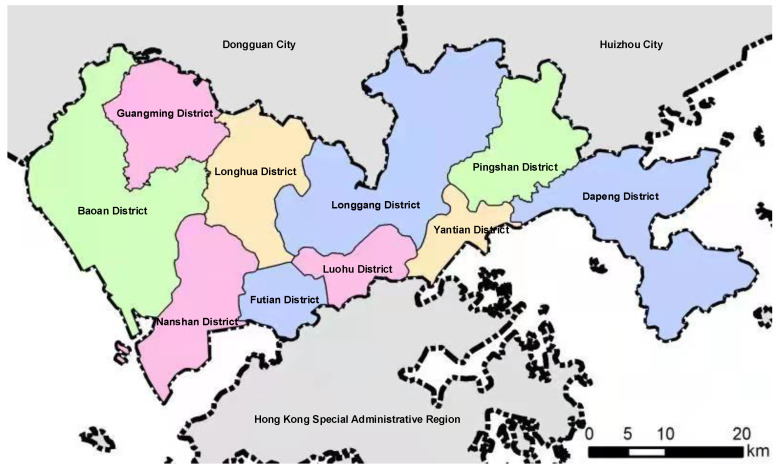
The location of Nanshan District.

**Figure 3 ijerph-19-00344-f003:**
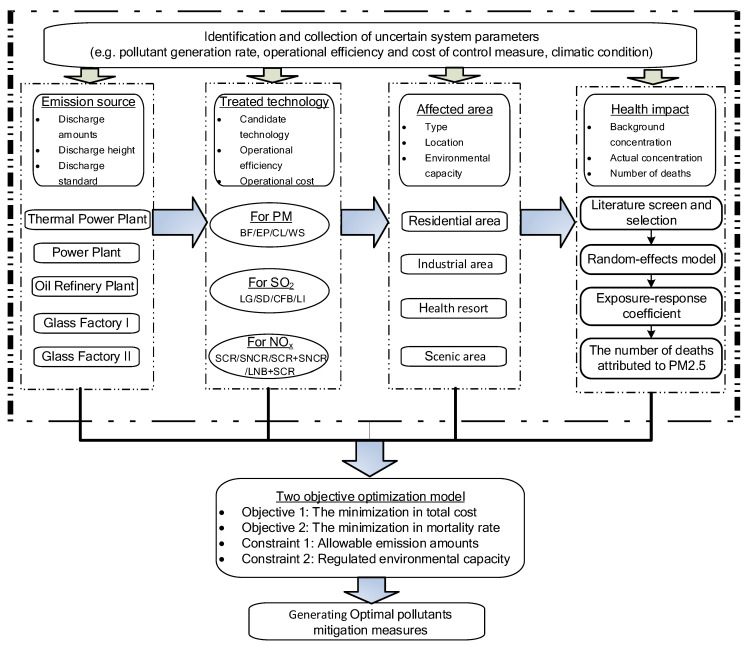
A schematic of the air quality management system in the Nanshan district.

**Figure 4 ijerph-19-00344-f004:**
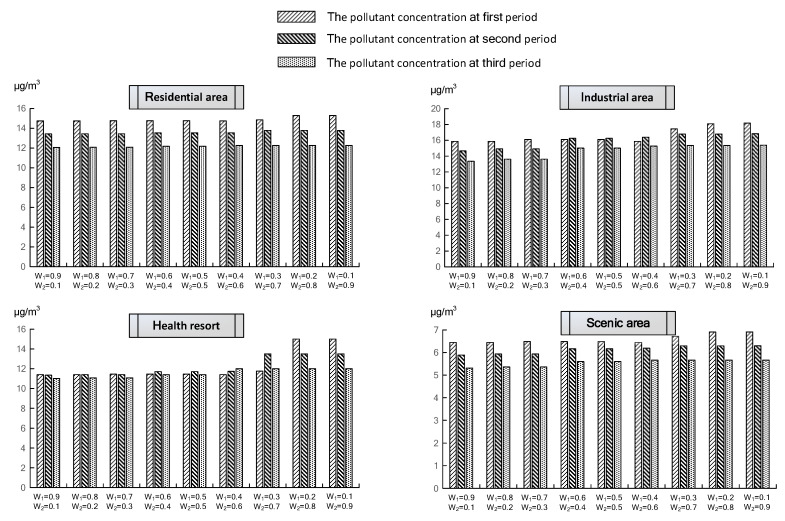
The residual PM_2.5_ concentration of four affected regions over three periods.

**Figure 5 ijerph-19-00344-f005:**
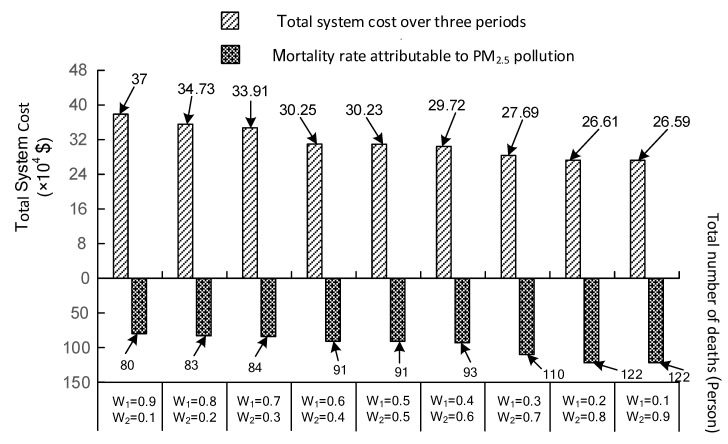
The comparison of two objective values under various weight combinations.

**Table 1 ijerph-19-00344-t001:** Information of 15 peer-reviewed articles with the relationship coefficient of exposure to reaction (i.e., β) used for the meta-analysis.

Serial Number of Included Literatures	Authors	Research Area	Published Period	*β*	95% CI
[57]	Yang et al.	Guangzhou	2012	0.009	(0.0055~0.0126)
[51]	Geng et al.	Shanghai	2013	0.0057	(0.0012~0.0101)
[50]	Chen et al.	Shanghai	2013	0.0017	(0.0002~0.0035)
[49]	Chen et al.	Shanghai	2011	0.0047	(0.0022~0.0079)
[48]	Chen et al.	Shanghai	2011	0.0047	(0.0022~0.0072)
[58]	Li et al.	Shanghai	2013	0.0043	(0.0014~0.0073)
[56]	Wu et al.	Guangzhou	2018	0.0055	(0.0024~0.0086)
[61]	Zhang et al	Shenzhen	2016	0.0069	(0.0055~0.0083)
[54]	Zhou et al	Fuzhou	2018	0.0017	(−0.0009~0.0043)
[52]	Feng et al	Changsha	2018	0.00518	(0.00065~0.00994)
[60]	Lin et al	Dongguan	2016	0.0052	(0.0024~0.008)
[60]	Lin et al	Foshan	2016	0.0091	(0.0061~0.0122)
[60]	Lin et al	Guangzhou	2016	0.0057	(0.0042~0.0073)
[60]	Lin et al	Jiangmen	2016	0.007	(0.0047~0.0093)
[60]	Lin et al	Shenzhen	2016	0.001	(−0.0004~0.0024)
[60]	Lin et al	Zhuhai	2016	0.0014	(−0.0006~0.0034)
[53]	Hu et al	Zhejiang Province	2018	0.0061	(0.0034~0.0089)
[62]	Li et al.	Pearl river delta	2017	0.0054	(0.0015~0.0092)
[55]	Zhu	Huizhou	2017	0.0095	(0.0013~0.0179)
[59]	Shi	Guangzhou	2015	0.012	(0.0063~0.0177)

Notes: CI = Confidence interval.

**Table 2 ijerph-19-00344-t002:** The parameter information related to the five emission sources.

Emission Sources	Average Discharge Height (m)	Pollutants	Discharge Amounts (t/d)		
			k = 1	k = 2	k = 3
Power plant Co. Ltd. (Shenzhen, China) (PPC)	50	PM	5.57	6.13	6.41
	50	SO_2_	69.31	76.24	79.71
	50	NO_x_	14.54	15.99	16.72
Power plant (Shenzhen, China) (PP)	210	PM	3.84	4.22	4.42
	210	SO_2_	47.70	52.47	54.86
	210	NO_x_	70.69	77.75	81.29
Oil Co. Ltd. (Shenzhen, China) (Oc)	20	PM	0.11	0.12	0.12
	20	SO_2_	0.10	0.11	0.12
	20	NO_x_	0.79	0.86	0.90
Glass Co. Ltd. 1 (Shenzhen, China) (Gc1)	120	PM	0.31	0.34	0.36
	120	SO_2_	1.62	1.78	1.86
	120	NO_x_	0.67	0.74	0.77
Glass Co. Ltd. 2 (Shenzhen, China) (Gc2)	30	PM	0.09	0.09	0.10
	30	SO_2_	0.08	0.09	0.09
	30	NO_x_	0.63	0.70	0.73

**Table 3 ijerph-19-00344-t003:** The treatment cost and efficiency of twelve candidate technologies.

Pollutants	Technologies	Indicators	Planning Period		
			k = 1	k = 2	k = 3
PM	Bag filter (BF)	OC	281.25	323.44	351.56
		TE	0.98	0.98	0.98
	Electrostatic precipitator (EP)	OC	173.28	199.27	216.60
		TE	0.93	0.93	0.93
	Cyclones (CL)	OC	54.69	62.89	68.36
		TE	0.65	0.65	0.65
	Wet scrubbers (WS)	OC	112.50	129.38	140.63
		TE	0.91	0.91	0.91
SO_2_	Limestone gypsum (LG)	OC	515.63	592.97	644.53
		TE	0.95	0.95	0.95
	Spray drying (SD)	OC	437.50	503.13	546.88
		TE	0.7	0.7	0.7
	Circulating fluid bed (CFB)	OC	343.75	395.31	429.69
		TE	0.9	0.9	0.9
	Limestone injection (LI)	OC	375.00	431.25	468.75
		TE	0.6	0.6	0.6
NO_x_	Selective Catalytic Reduction (SCR)	OC	225.00	258.75	281.25
		TE	0.8	0.8	0.8
	Selective non-catalytic reduction (SNCR)	OC	340	391	425
		TE	0.5	0.5	0.5
	SCR + SNCR	OC	312.50	359.38	390.63
		TE	0.75	0.75	0.75
	Low nitrogen burning (LNB) + SCR	OC	410.94	472.58	513.67
		TE	0.94	0.94	0.94

Notes: OC = Operational costs, USD/t; TE = Treatment efficiency, %.

**Table 4 ijerph-19-00344-t004:** The optimal treated PM magnitude of various techniques.

ES	T	w_1_ = 0.7 and w_2_ = 0.3	w_1_ = 0.6 and w_2_ = 0.4	w_1_ = 0.4 and w_2_ = 0.6	w_1_ = 0.3 and w_2_ = 0.7
PPc	k = 1	BF (5.57)	BF (5.57)	BF (5.57)	BF (5.57)
	k = 2	BF (6.13)	BF (6.13)	BF (6.13)	BF (6.13)
	k = 3	BF (6.41)	BF (6.41)	BF (6.41)	BF (6.41)
	Sum	BF (18.11)	BF (18.11)	BF (18.11)	BF (18.11)
PP	k = 1	BF (3.84)	BF (3.84)	WS (3.84)	WS (3.84)
	k = 2	BF (4.22)	BF (4.22)	WS (4.22)	WS (4.22)
	k = 3	BF (4.42)	BF (4.42)	WS (4.42)	WS (4.42)
	Sum	BF (12.48)	BF (12.48)	WS (12.48)	WS (12.48)
Oc	k = 1	BF (0.11)	WS (0.11)	WS (0.11)	WS (0.11)
	k = 2	BF (0.12)	WS (0.12)	WS (0.12)	WS (0.12)
	k = 3	WS (0.12)	WS (0.12)	WS (0.12)	WS (0.12)
	Sum	BF (0.23) WS (0.12)	WS (0.35)	WS (0.35)	WS (0.35)
Gc1	k = 1	BF (0.31)	BF (0.31)	BF (0.31)	BF (0.31)
	k = 2	BF (0.34)	BF (0.34)	BF (0.34)	BF (0.34)
	k = 3	BF (0.36)	BF (0.36)	BF (0.36)	BF (0.36)
	Sum	BF (1.01)	BF (1.01)	BF (1.01)	BF (1.01)
Gc2	k = 1	BF (0.09)	BF (0.09)	WS (0.09)	WS (0.09)
	k = 2	BF (0.09)	BF (0.09)	WS (0.09)	WS (0.09)
	k = 3	BF (0.10)	BF (0.10)	WS (0.10)	WS (0.10)
	Sum	BF (0.28)	BF (0.28)	WS (0.28)	WS (0.28)

Notes: ES = emission sources, where the abbreviations of emission source and candidate technology are consistent with those in Table 2 and Table 3, respectively. The number inside parentheses represents the treated amounts (t/d) of relevant technology.

**Table 5 ijerph-19-00344-t005:** The optimal treated SO_2_ magnitude of various techniques.

ES	T	w_1_ = 0.7 and w_2_ = 0.3	w_1_ = 0.6 and w_2_ = 0.4	w_1_ = 0.4 and w_2_ = 0.6	w_1_ = 0.3 and w_2_ = 0.7
*PPc*	k = 1	LG (69.31)	LG (69.31)	LG (69.31)	LG (69.31)
	k = 2	LG (76.24)	LG (76.24)	LG (76.24)	LG (42.17) CFB (34.07)
	k = 3	LG (79.71)	LG (79.71)	LG (68.55) CFB (11.16)	LG (68.69) CFB (11.02)
	Sum	LG (225.26)	LG (225.26)	LG (214.10) CFB (11.16)	LG (180.17) CFB (45.09)
*PP*	k = 1	CFB (47.70)	CFB (47.70)	CFB (47.70)	CFB (47.70)
	k = 2	CFB (52.47)	CFB (52.47)	CFB (52.47)	CFB (52.47)
	k = 3	CFB (54.86)	CFB (54.86)	CFB (54.86)	CFB (54.86)
	Sum	CFB (155.03)	CFB (155.03)	CFB (155.03)	CFB (155.03)
*Oc*	k = 1	CFB (0.10)	CFB (0.10)	CFB (0.10)	CFB (0.10)
	k = 2	CFB (0.11)	CFB (0.11)	CFB (0.11)	CFB (0.11)
	k = 3	CFB (0.12)	CFB (0.12)	CFB (0.12)	CFB (0.12)
	Sum	CFB (0.33)	CFB (0.33)	CFB (0.33)	CFB (0.33)
*Gc1*	k = 1	LG (1.62)	LG (1.62)	CFB (1.62)	CFB (1.62)
	k = 2	LG (1.78)	CFB (1.78)	CFB (1.78)	CFB (1.78)
	k = 3	LG (1.86)	CFB (1.86)	CFB (1.86)	CFB (1.86)
	Sum	LG (5.26)	LG (1.62) CFB (3.64)	CFB (5.26)	CFB (5.26)
*Gc2*	k = 1	CFB (0.08)	CFB (0.08)	CFB (0.08)	CFB (0.08)
	k = 2	CFB (0.09)	CFB (0.09)	CFB (0.09)	CFB (0.09)
	k = 3	CFB (0.09)	CFB (0.09)	CFB (0.09)	CFB (0.09)
	Sum	CFB (0.26)	CFB (0.26)	CFB (0.26)	CFB (0.26)

Notes: ES = emission sources, where the abbreviations of emission source and candidate technology are consistent with those in Table 2 and Table 3, respectively. The number inside parentheses represents the treated amounts (t/d) of relevant technology.

**Table 6 ijerph-19-00344-t006:** The optimal treated NO_X_ magnitude of various techniques.

ES	T	w_1_ = 0.7 and w_2_ = 0.3	w_1_ = 0.6 and w_2_ = 0.4	w_1_ = 0.4 and w_2_ = 0.6	w_1_ = 0.3 and w_2_ = 0.7
PPc	k = 1	LNB + SCR (14.54)	LNB + SCR (14.54)	LNB + SCR (14.54)	LNB + SCR (14.54)
	k = 2	LNB + SCR (15.99)	LNB + SCR (15.99)	LNB + SCR (15.99)	LNB + SCR (15.99)
	k = 3	LNB + SCR (16.72)	LNB + SCR (16.72)	LNB + SCR (16.72)	LNB + SCR (16.72)
	Sum	LNB + SCR (47.25)	LNB + SCR (47.25)	LNB + SCR (47.25)	LNB + SCR (47.25)
PP	k = 1	LNB + SCR (70.69)	LNB + SCR (70.69)	LNB + SCR (70.69)	SCR (70.69)
	k = 2	LNB + SCR (77.75)	SCR (77.75)	SCR (77.75)	SCR (77.75)
	k = 3	LNB + SCR (81.29)	SCR (81.29)	SCR (81.29)	SCR (81.29)
	Sum	LNB + SCR (229.73)	SCR (159.04) LNB + SCR (70.69)	SCR (159.04) LNB + SCR (70.69)	SCR (229.73)
Oc	k = 1	LNB + SCR (0.79)	LNB + SCR (0.79)	SCR (0.79)	SCR (0.79)
	k = 2	SCR (0.86)	SCR (0.86)	SCR (0.86)	SCR (0.86)
	k = 3	SCR (0.90)	SCR (0.90)	SCR (0.90)	SCR (0.90)
	Sum	SCR (1.76) LNB + SCR (0.79)	SCR (1.76) LNB + SCR (0.79)	SCR (2.55)	SCR (2.55)
Gc1	k = 1	LNB + SCR (0.67)	LNB + SCR (0.67)	LNB + SCR (0.67)	LNB + SCR (0.67)
	k = 2	LNB + SCR (0.74)	LNB + SCR (0.74)	LNB + SCR (0.74)	SCR (0.74)
	k = 3	LNB + SCR (0.77)	LNB + SCR (0.77)	LNB + SCR (0.77)	SCR (0.77)
	Sum	LNB + SCR (2.18)	LNB + SCR (2.18)	LNB + SCR (2.18)	SCR (1.51) LNB + SCR (0.67)
Gc2	k = 1	LNB + SCR (0.63)	LNB + SCR (0.63)	LNB + SCR (0.63)	SCR (0.63)
	k = 2	LNB + SCR (0.70)	SCR (0.70)	SCR (0.70)	SCR (0.70)
	k = 3	LNB + SCR (0.73)	SCR (0.73)	SCR (0.73)	SCR (0.73)
	Sum	LNB + SCR (2.06)	SCR (1.43) LNB + SCR (0.63)	SCR (1.43) LNB + SCR (0.63)	SCR (2.06)

Notes: ES = emission sources, where the abbreviations of emission source and candidate technology are consistent with those in Table 2 and Table 3, respectively. The number inside parentheses represents the treated amounts (t/d) of relevant technology.

## Data Availability

Not applicable.

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
