# Peer review of "An Integration Method for Regional PM_2.5_ Pollution Control Optimization Based on Meta-Analysis and Systematic Review"

_ijerph, 2021, doi:10.3390/ijerph19010344_

Round 1

Reviewer 1 Report

The study proposes a two-objective optimization model to formulate an effective air pollution control strategy, taking into account diverse parameters, including health impact assessment. It brings novelty to the subject field and has the potential for publication on IJERPH.

However, a major review is needed.

  • The data sources are not clear. Please improve the writing about it.
  • The authors should give more details about the methodology, such as:
    • How were the results from the Gaussian Diffusion model included? As it comprises a future scenario, which meteorological database did you use on the Gaussian Diffusion model? Maybe the authors should address some more details about the appliance of the Gaussian Diffusion model and the other variables in an Appendix.
    • Which values were used for the pollutants conversion ratio for PM2.5, based on which references?
  • Why did you use TSP, NOx and SO2 as emissions to account for PM2.5 emissions? You need to explain. Why was NH3 not considered, as it is also a PM2.5 precursor? You need to explain your choice, based on the literature.
  • Line 282. Before Figure 1, you should call it on the text.
  • Please include a map showing the Nanshan District location.
  • Please define RMB. It is common in China but not worldwide. If possible, please convert the values for the US dollar, as it is a more common measure.
  • Line 290, please change for “…total area of 192 km2, a total population of 1.5 million, and gross domestic product of 500 billion RMB…”.
  • I did not understand the real meaning of k=1, k=2 and k=3 in Table 2. Is it each of the selected years (2020-2022)? If so, you should change Line 337 as follows: “…divided into three planning periods of one year each (k=1 for 2020, k=2 for 2021 and k=3 for 2022).
  • Also, it is not clear what the planning period measure unity on table 2, is it tons a day?
  • Tables 4 to 6 are hard to understand. I suggest using the following abbreviations for the sources, instead of S1 to S5, and also use them in Table 2:

Power plant Co. Ltd (PPc)

Power plant (PP)

Oil Co. Ltd (Oc)

Glass Co. Ltd 1 (Gc1)

Glass Co. Ltd 2 (Gc2)

  • In tables 4 to 6, instead of using 1 to 4 for the treatment technologies, please use the abbreviations used in Table 3. BF for Bag Filter, EP for Electrostatic precipitator, and so on.
  • To improve the understanding of the results, I also suggest you change Tables 4 to 6. As an example, see below for Table 4, the S3 emission source.

ES

T

w1=0.7 and w2=0.3

Oc

K=1

BF (0.11)

K=2

BF (0.12)

K=3

WS (0.12)

Sum

BF (0.23)

WS (0.12)

It would simplify your results. This full of zero tables confused me.

  • Page 17, Line 3, the unit t/a, is tons per??
  • Page 17, line 43. Unbold “Figure 3”.
  • You need to include the legend of Figure 3 y-axis.
  • English spelling minor review.

Author Response

RESPONSES TO REVIEWERS’ COMMENTS

We are deeply grateful to the reviewers for their insightful reviews. The provided comments and suggestions have contributed much to improving the manuscript. According to them, we have made efforts to revise the manuscript, with the details explained as follows:

Note: For the reviewer’s convenience of re-review, we divided his/her comments into several parts with ordinal numbers, and corresponded to them in the following responses.

Responses to Reviewer #1’s comments

COMMENT 1 -- (1) The data sources are not clear. Please improve the writing about it.

RESPONSE: We much appreciate the reviewer’s comments. The various types of data were obtained based on specific data-acquisition method. Firstly, the meta-analysis method was used to estimate the percentage of deaths caused by PM2.5 pollution through summarizing and combining the research outputs on the same subject. Next, the parameters involved in the calculation of mortality caused by PM2.5 pollution were sourced from several information channel. For example, the total number of deaths was collected from local statistical yearbook. The reference concentration of PM2.5 was determined based on the investigation in natural background value and the hygienic standards established by government agencies. The conversion ratio between precursor and PM2.5 was designed based on chemical composition analysis and source apportionment results provided by local environmental protection agency and study output of relevant research. Finally, the variables included in the optimization model were collected based on several ways. For instance, the related information of pollutants sources, including their location, heights and discharged pollutants magnitude, was obtained through on-site survey. Local available pollutants-mitigation technologies with their treated cost and efficiency were determined based on field investigation, literature review and expert consultation. For the discharge standard of pollutants sources and environmental load capacity of affected areas, they are designated according to the emission-discharge standards and air quality standards released by environmental protection agency. Moreover, we have clarified the raised issue as follows (on Pages 6, 7, 12 and 13):

“By referring to the Shenzhen Nanshan District statistical yearbook, Shenzhen Health Statistical Yearbook, and Shenzhen Statistical Yearbook, particularly those belonging to the year 2018, the latest resident mortality data (including the total number of deaths and average mortality rate) of Shenzhen residents were obtained.

Besides, the reference concentrations available for Shenzhen include the natural background concentrations of air pollutants, the minimum concentration observed in the past epidemiological literatures, and the hygienic standards established by government agencies.  

The above three conversion ratios were determined based on chemical composition analysis and source apportionment results provided by local environmental protection agency and study result of relevant research (Zhang et al., 2018; Liu et al., 2019; Wu et al., 2020; Yang et al., 2020).

The on-site survey and statistical results provided by EIA (Environmental Impact Assessment) reports demonstrated that the pollutants discharged from the above-mentioned five pollutant sources mainly included PM, SO2 and NOx, where they are considered as the precursor pollutants that contributed to PM2.5.

Table 2 provides the relevant information of five sources, which are mostly obtained from on-site investigation and data statistics. Moreover, their discharge standards were regulated by environmental protection agency based on the types of industries.

The treatment cost and efficiency of candidate technologies (as shown in Table 3) were mainly determined based on the field investigation, literature review and expert consultation.

In this study, the annual mean concentration of PM2.5 regulated by the Ambient Air Quality Standards released by China (GB3095-2012) was used as a reference.”

COMMENT 2 -- (2) How were the results from the Gaussian Diffusion model included? As it comprises a future scenario, which meteorological database did you use on the Gaussian Diffusion model? Maybe the authors should address some more details about the appliance of the Gaussian Diffusion model and the other variables in an Appendix.

RESPONSE: We much appreciate the reviewer’s comments. In this study, the Gaussian Diffusion model is mainly used to calculate the PM2.5 concentration of the affected areas based on the treatment efficiencies of control technologies and precursors’ conversion ratios to PM2.5. The generated result of Gaussian Diffusion model was incorporated into the two-objective optimization model for identifying optimum pollutant mitigation measure, as reflected in the objective function (2a) and constraint (2i). Considering the planning period (2020-2022) is relatively short, it is assumed that the meteorological parameters and related information of pollutants sources (i.e. their location and discharge height) remain unchanged. Moreover, we have elaborated the Gaussian Diffusion model as follows in the Appendix:

“Appendix I - Gaussian diffusion model

The atmospheric diffusion model

The atmospheric diffusion model is a physical and mathematical model for reflecting the transport and diffusion process of air pollutants in the atmosphere. Due to the flat cushion surface in Nanshan District, the pollutants diffusion almost obeys the normal distribution and the distance among the pollutant sources and affected areas is less than ten kilometers, thus, as the most commonly used diffusion model, the Gaussian diffusion model was selected to simulate the continuous point source diffusion of the flat terrain elevation and calculate the PM2.5 concentration in affected areas.

The determination of major parameters

The ground concentration model of Gaussian diffusion mode is suitable to calculate the maximum ground concentration of air pollutant, where the critical step is to determine the effective stack height and atmospheric diffusion parameters.

The calculation of effective stack height

(A-1)

where He is the effective stack height (m); h is the geometric height of stack (m);  is the plume rise height (m), which is obtained based on the equation (A-2).

(A-2)

where n0 is the coefficient of plume thermal condition and surface condition; Qh is the thermal emission rate of exhaust gas (kJ/s); n1 is the heat release rate of plume; n2 is the height index of discharge source; Uh is the wind speed at the height h (m/s), which is calculated as follows:

(A-3)

where U10 is the wind speed at the height of 10 m (m/s); p is the index.  

The estimation of atmospheric diffusion parameters

The atmospheric diffusion parameters are the function of downwind distance, atmospheric stability and ground roughness, as described by two equations (A-4) and (A-5), which are formulated as follows:

(A-4)

(A-5)

where σy is the transverse diffusion parameter perpendicular to the dominant wind direction; σz is the vertical diffusion parameter; x is the distance between pollutants source and affected area (m); γ1, γ2, α1 and α2 are empirical regression coefficients at distance x.

The calculation of ground concentration at normal wind speed

Considering the ground position of the pollutant sources as the origin, the downwind ground concentration at the distance x is calculated as follows:

(A-6)

where C is the ground concentration at the distance x, mg/m3; Q is discharged magnitude of pollutants (mg/s).”

COMMENT 3 -- (3) Which values were used for the pollutants conversion ratio for PM2.5, based on which references?

RESPONSE: We much appreciate the reviewer’s comments, and have clarified the raised issue as follows (on Page 7):

“The above three conversion ratios were determined from chemical composition analysis and source apportionment results provided by local environmental protection agency and study output of relevant research works (Zhang et al., 2018; Liu et al., 2019; Wu et al., 2020; Yang et al., 2020).”

References:

“Liu, Y.C., Wu, Z.J., Huang, X.F., Shen, H.Y., Bai, Y., Qiao, K., Meng, X.X.Y., Hu, W.W., Tang, M.J., He, L.Y. 2019. Aerosol phase state and its link to chemical composition and liquid water content in a subtropical coastal megacity. ENVIRONMENTAL SCIENCE & TECHNOLOGY, 53, 5027-5033.

Wu, L., Wang, Y., Li, L., Zhang, G. 2020. Acidity and inorganic ion formation in PM2.5 based on continuous online observations in a South China megacity. ATMOSPHERIC POLLUTION RESEARCH, 11, 1339-1350.

Yang, H.L., Zhang, Y., Li, L., Wai, C.P., Lu, C., Zhang, L. 2020. Characteristics of aerosol pollution under different visibility conditions in winter in a coastal mega-city in China. JOURNAL OF TROPICAL METEOROLOGY, 26, 231-238.

Zhang, R., Sun, X.S., Shi, A.J., Huang, Y.H., Yan, J., Nie, T., Yan, X., Li, X. 2018. Secondary inorganic aerosols formation during haze episodes at an urban site in Beijing, China. ATMOSPHERIC ENVIRONMENT 177, 275-282.” 

We also checked this reference carefully and made all necessary corrections.

COMMENT 4 -- (4) Why did you use PM, NOx and SO2 as emissions to account for PM2.5 emissions? You need to explain. Why was NH3 not considered, as it is also a PM2.5 precursor? You need to explain your choice, based on the literature.

RESPONSE: We much appreciate the reviewer’s comments. As a major precursor, the pollutant NH3 indeed should be considered as the control target. However, the investigated results of the five major emission sources demonstrated that only three major types of pollutants (including PM, SO2 and NOx) were involved. Therefore, the contribution of NH3 to PM2.5 was ignored in this study. To clarify this concern, we have added the following explanation in the revised manuscript (on Page 12):

“The on-site survey and statistical results provided by EIA (Environmental Impact Assessment) reports demonstrated that the pollutants discharged from the above-mentioned five sources mainly included PM, SO2 and NOx, where they are considered as the precursor pollutants that contributed to PM2.5.”

COMMENT 5 -- (5) Line 282. Before Figure 1, you should call it on the text.

RESPONSE: We much appreciate the reviewer’s comments, and have clarified the raised issue as follows (on Page 8):

“The overall framework of the model is shown in Figure 1.”

COMMENT 6 -- (6) Please include a map showing the Nanshan District location.

RESPONSE: We accept the reviewer’s suggestion, and have added Figure 2 as follows (on Page 10):

Figure 2. The location of Nanshan District.

COMMENT 7 -- (7) Please define RMB. It is common in China but not worldwide. If possible, please convert the values for the US dollar, as it is a more common measure.

RESPONSE: We accept the reviewer’s suggestion and have converted the unit RMB into the unit $ in the text, table and figure of revised manuscript as follows (on Pages 2, 9, 13, 18 and 19):

“The resulting Disability-Adjusted Life-Years (DALYs) loss in China amounted to 16.09 million $ or 4.2 percent of the global DALYs loss.

Nanshan district has a total area of 192 km2, a total population of 1.5 million, and gross domestic product of 78.13 billion $, respectively, in 2019.

Conversely, the total system cost decreases from 37 to 26.58 ×104 $, respectively.”

Table 3. The treatment cost and efficiency of twelve candidate technologies.

Pollutants

Technologies

indicators

Planning period

k = 1

k = 2

k = 3

PM

Bag filter (BF)

OC

281.25

323.44

351.56

TE

0.98

0.98

0.98

Electrostatic precipitator (EP)

OC

173.28

199.27

216.60

TE

0.93

0.93

0.93

Cyclones (CL)

OC

54.69

62.89

68.36

TE

0.65

0.65

0.65

Wet scrubbers (WS)

OC

112.50

129.38

140.63

TE

0.91

0.91

0.91

SO2

Limestone gypsum (LG)

OC

515.63

592.97

644.53

TE

0.95

0.95

0.95

Spray drying (SD)

OC

437.50

503.13

546.88

TE

0.7

0.7

0.7

Circulating fluid bed (CFB)

OC

343.75

395.31

429.69

TE

0.9

0.9

0.9

Limestone injection (LI)

OC

375.00

431.25

468.75

TE

0.6

0.6

0.6

NOx

Selective Catalytic Reduction (SCR)

OC

225.00

258.75

281.25

TE

0.8

0.8

0.8

Selective non-catalytic reduction (SNCR)

OC

340

391

425

TE

0.5

0.5

0.5

SCR+SNCR

OC

312.50

359.38

390.63

TE

0.75

0.75

0.75

Low nitrogen burning (LNB) + SCR

OC

410.94

472.58

513.67

TE

0.94

0.94

0.94

Notes: OC = Operational costs, $/t; TE = Treatment efficiency, %.

Figure 5. The comparison of two objective values under various weight combinations.

COMMENT 8 -- (8) Line 290, please change for “…total area of 192 km2, a total population of 1.5 million, and gross domestic product of 500 billion RMB…”.

RESPONSE: We accept the reviewer’s suggestion, and have made the revision as follows (on Page 9):

“Nanshan district has a total area of 192 km2, a total population of 1.5 million, and gross domestic product of 78.13 billion $ in 2019.”

COMMENT 9 -- (9) I did not understand the real meaning of k=1, k=2 and k=3 in Table 2. Is it each of the selected years (2020-2022)? If so, you should change Line 337 as follows: “…divided into three planning periods of one year each (k=1 for 2020, k=2 for 2021 and k=3 for 2022).

RESPONSE: We accept the reviewer’s suggestion and have made the correction in the revised manuscript as follows (on Page 12):  

“The time scope was set to three years (2020-2022), which was further split into three planning periods with each period including one year (i.e. k=1 for 2020, k=2 for 2021 and k=3 for 2022).”

COMMENT 10 -- (10) Also, it is not clear what the planning period measure unity on table 2, is it tons a day?

RESPONSE: We appreciate the reviewer’s carefulness. Accordingly, we have corrected the above errors as follows (on Page 13):

Table 2. The parameter information related to the five emission sources

Emission sources

Average discharge height(m)

Pollutants

Discharge amounts (t/d)

k = 1

k = 2

k = 3

Power plant Co. Ltd (PPC)

50

PM

5.57

6.13

6.41

50

SO2

69.31

76.24

79.71

50

NOx

14.54

15.99

16.72

Power plant

(PP)

210

PM

3.84

4.22

4.42

210

SO2

47.70

52.47

54.86

210

NOx

70.69

77.75

81.29

Oil Co. Ltd

(Oc)

20

PM

0.11

0.12

0.12

20

SO2

0.10

0.11

0.12

20

NOx

0.79

0.86

0.90

Glass Co. Ltd 1

(Gc1)

120

PM

0.31

0.34

0.36

120

SO2

1.62

1.78

1.86

120

NOx

0.67

0.74

0.77

Glass Co. Ltd 2

(Gc2)

30

PM

0.09

0.09

0.10

30

SO2

0.08

0.09

0.09

30

NOx

0.63

0.70

0.73

COMMENT 11 -- (11) Tables 4 to 6 are hard to understand. I suggest using the following abbreviations for the sources, instead of S1 to S5, and also use them in Table 2: Power plant Co. Ltd (PPc), Power plant (PP), Oil Co. Ltd (Oc), Glass Co. Ltd 1 (Gc1) and Glass Co. Ltd 2 (Gc2).

RESPONSE: We accept the reviewer’s suggestion, and have revised the Tables 2, 4, 5 and 6 as follows (on Pages 12, 14, 15 and 16):

Table 2. The parameter information related to the five emission sources

Emission sources

Average discharge height(m)

Pollutants

Discharge amounts (t/d)

k = 1

k = 2

k = 3

Power plant Co. Ltd (PPC)

50

PM

5.57

6.13

6.41

50

SO2

69.31

76.24

79.71

50

NOx

14.54

15.99

16.72

Power plant

(PP)

210

PM

3.84

4.22

4.42

210

SO2

47.70

52.47

54.86

210

NOx

70.69

77.75

81.29

Oil Co. Ltd

(Oc)

20

PM

0.11

0.12

0.12

20

SO2

0.10

0.11

0.12

20

NOx

0.79

0.86

0.90

Glass Co. Ltd 1

(Gc1)

120

PM

0.31

0.34

0.36

120

SO2

1.62

1.78

1.86

120

NOx

0.67

0.74

0.77

Glass Co. Ltd 2

(Gc2)

30

PM

0.09

0.09

0.10

30

SO2

0.08

0.09

0.09

30

NOx

0.63

0.70

0.73

Table 4. The optimal treated PM magnitude of various techniques

ES

T

w1 = 0.7 and w2 = 0.3

w1 = 0.6 and w2 = 0.4

w1 = 0.4 and w2 = 0.6

w1 = 0.3 and w2 = 0.7

PPc

k = 1

BF(5.57)

BF(5.57)

BF(5.57)

BF(5.57)

k = 2

BF (6.13)

BF (6.13)

BF (6.13)

BF (6.13)

k = 3

BF(6.41)

BF(6.41)

BF(6.41)

BF(6.41)

Sum

BF(18.11)

BF(18.11)

BF(18.11)

BF(18.11)

PP

k = 1

BF(3.84)

BF(3.84)

WS(3.84)

WS(3.84)

k = 2

BF(4.22)

BF(4.22)

WS(4.22)

WS(4.22)

k = 3

BF(4.42)

BF(4.42)

WS(4.42)

WS(4.42)

Sum

BF(12.48)

BF(12.48)

WS(12.48)

WS(12.48)

Oc

k = 1

BF(0.11)

WS(0.11)

WS(0.11)

WS(0.11)

k = 2

BF(0.12)

WS(0.12)

WS(0.12)

WS(0.12)

k = 3

WS(0.12)

WS(0.12)

WS(0.12)

WS(0.12)

Sum

BF(0.23)

WS(0.12)

WS(0.35)

WS(0.35)

WS(0.35)

Gc1

k = 1

BF(0.31)

BF(0.31)

BF(0.31)

BF(0.31)

k = 2

BF(0.34)

BF(0.34)

BF(0.34)

BF(0.34)

k = 3

BF(0.36)

BF(0.36)

BF(0.36)

BF(0.36)

Sum

BF(1.01)

BF(1.01)

BF(1.01)

BF(1.01)

Gc2

k = 1

BF(0.09)

BF(0.09)

WS(0.09)

WS(0.09)

k = 2

BF(0.09)

BF(0.09)

WS(0.09)

WS(0.09)

k = 3

BF(0.10)

BF(0.10)

WS(0.10)

WS(0.10)

Sum

BF(0.28)

BF(0.28)

WS(0.28)

WS(0.28)

Notes: ES = emission sources, where the abbreviations of emission source and candidate technology are consistent with those in Table 2 and Table 3, respectively. The number inside parentheses represents the treated amounts (t/d) of relevant technology.

Table 5. The optimal treated SO2 magnitude of various techniques

ES

T

w1 = 0.7 and w2 = 0.3

w1 = 0.6 and w2 = 0.4

w1 = 0.4 and w2 = 0.6

w1 = 0.3 and w2 = 0.7

PPc

k = 1

LG(69.31)

LG(69.31)

LG(69.31)

LG(69.31)

k = 2

LG(76.24)

LG(76.24)

LG(76.24)

LG(42.17)

CFB(34.07)

k = 3

LG(79.71)

LG(79.71)

LG(68.55)

CFB(11.16)

LG(68.69)

CFB(11.02)

Sum

LG(225.26)

LG(225.26)

LG(214.10)

CFB(11.16)

LG(180.17)

CFB(45.09)

PP

k = 1

CFB(47.70)

CFB(47.70)

CFB(47.70)

CFB(47.70)

k = 2

CFB(52.47)

CFB(52.47)

CFB(52.47)

CFB(52.47)

k = 3

CFB(54.86)

CFB(54.86)

CFB(54.86)

CFB(54.86)

Sum

CFB(155.03)

CFB(155.03)

CFB(155.03)

CFB(155.03)

Oc

k = 1

CFB(0.10)

CFB(0.10)

CFB(0.10)

CFB(0.10)

k = 2

CFB(0.11)

CFB(0.11)

CFB(0.11)

CFB(0.11)

k = 3

CFB(0.12)

CFB(0.12)

CFB(0.12)

CFB(0.12)

Sum

CFB(0.33)

CFB(0.33)

CFB(0.33)

CFB(0.33)

Gc1

k = 1

LG(1.62)

LG(1.62)

CFB(1.62)

CFB(1.62)

k = 2

LG(1.78)

CFB(1.78)

CFB(1.78)

CFB(1.78)

k = 3

LG(1.86)

CFB (1.86)

CFB (1.86)

CFB (1.86)

Sum

LG(5.26)

LG(1.62)

CFB(3.64)

CFB(5.26)

CFB(5.26)

Gc2

k = 1

CFB(0.08)

CFB(0.08)

CFB(0.08)

CFB(0.08)

k = 2

CFB(0.09)

CFB(0.09)

CFB(0.09)

CFB(0.09)

k = 3

CFB(0.09)

CFB(0.09)

CFB(0.09)

CFB(0.09)

Sum

CFB(0.26)

CFB(0.26)

CFB(0.26)

CFB(0.26)

Notes: ES = emission sources, where the abbreviations of emission source and candidate technology are consistent with those in Table 2 and Table 3, respectively. The number inside parentheses represents the treated amounts (t/d) of relevant technology.

Table 6. The optimal treated NOX magnitude of various techniques

ES

T

w1 = 0.7 and w2 = 0.3

w1 = 0.6 and w2 = 0.4

w1 = 0.4 and w2 = 0.6

w1 = 0.3 and w2 = 0.7

PPc

k = 1

LNB+SCR(14.54)

LNB+SCR(14.54)

LNB+SCR(14.54)

LNB+SCR(14.54)

k = 2

LNB+SCR(15.99)

LNB+SCR(15.99)

LNB+SCR(15.99)

LNB+SCR(15.99)

k = 3

LNB+SCR(16.72)

LNB+SCR(16.72)

LNB+SCR(16.72)

LNB+SCR(16.72)

Sum

LNB+SCR(47.25)

LNB+SCR(47.25)

LNB+SCR(47.25)

LNB+SCR(47.25)

PP

k = 1

LNB+SCR(70.69)

LNB+SCR(70.69)

LNB+SCR(70.69)

SCR(70.69)

k = 2

LNB+SCR(77.75)

SCR(77.75)

SCR(77.75)

SCR(77.75)

k = 3

LNB+SCR(81.29)

SCR(81.29)

SCR(81.29)

SCR(81.29)

Sum

LNB+SCR(229.73)

SCR(159.04) LNB+SCR(70.69)

SCR(159.04) LNB+SCR(70.69)

SCR(229.73)

Oc

k = 1

LNB+SCR(0.79)

LNB+SCR(0.79)

SCR(0.79)

SCR(0.79)

k = 2

SCR(0.86)

SCR(0.86)

SCR(0.86)

SCR(0.86)

k = 3

SCR(0.90)

SCR(0.90)

SCR(0.90)

SCR(0.90)

Sum

SCR(1.76)

LNB+SCR(0.79)

SCR(1.76)

LNB+SCR(0.79)

SCR(2.55)

SCR(2.55)

Gc1

k = 1

LNB+SCR(0.67)

LNB+SCR(0.67)

LNB+SCR(0.67)

LNB+SCR(0.67)

k = 2

LNB+SCR(0.74)

LNB+SCR(0.74)

LNB+SCR(0.74)

SCR(0.74)

k = 3

LNB+SCR(0.77)

LNB+SCR(0.77)

LNB+SCR(0.77)

SCR(0.77)

Sum

LNB+SCR(2.18)

LNB+SCR(2.18)

LNB+SCR(2.18)

SCR(1.51)

LNB+SCR(0.67)

Gc2

k = 1

LNB+SCR(0.63)

LNB+SCR(0.63)

LNB+SCR(0.63)

SCR(0.63)

k = 2

LNB+SCR(0.70)

SCR(0.70)

SCR(0.70)

SCR(0.70)

k = 3

LNB+SCR(0.73)

SCR(0.73)

SCR(0.73)

SCR(0.73)

Sum

LNB+SCR(2.06)

SCR(1.43)

LNB+SCR(0.63)

SCR(1.43)

LNB+SCR(0.63)

SCR(2.06)

Notes: ES = emission sources, where the abbreviations of emission source and candidate technology are consistent with those in Table 2 and Table 3, respectively. The number inside parentheses represents the treated amounts (t/d) of relevant technology.

COMMENT 12 -- (12) To improve the understanding of the results, I also suggest you change Tables 4 to 6. As an example, see below for Table 4, the S3 emission source. It would simplify your results. This full of zero tables confused me.

RESPONSE: We accept the reviewer’s suggestion, and have revised the Tables 4 to 6 as follows (on Pages 14 to 16):

Table 4. The optimal treated PM magnitude of various techniques

ES

T

w1 = 0.7 and w2 = 0.3

w1 = 0.6 and w2 = 0.4

w1 = 0.4 and w2 = 0.6

w1 = 0.3 and w2 = 0.7

PPc

k = 1

BF(5.57)

BF(5.57)

BF(5.57)

BF(5.57)

k = 2

BF (6.13)

BF (6.13)

BF (6.13)

BF (6.13)

k = 3

BF(6.41)

BF(6.41)

BF(6.41)

BF(6.41)

Sum

BF(18.11)

BF(18.11)

BF(18.11)

BF(18.11)

PP

k = 1

BF(3.84)

BF(3.84)

WS(3.84)

WS(3.84)

k = 2

BF(4.22)

BF(4.22)

WS(4.22)

WS(4.22)

k = 3

BF(4.42)

BF(4.42)

WS(4.42)

WS(4.42)

Sum

BF(12.48)

BF(12.48)

WS(12.48)

WS(12.48)

Oc

k = 1

BF(0.11)

WS(0.11)

WS(0.11)

WS(0.11)

k = 2

BF(0.12)

WS(0.12)

WS(0.12)

WS(0.12)

k = 3

WS(0.12)

WS(0.12)

WS(0.12)

WS(0.12)

Sum

BF(0.23)

WS(0.12)

WS(0.35)

WS(0.35)

WS(0.35)

Gc1

k = 1

BF(0.31)

BF(0.31)

BF(0.31)

BF(0.31)

k = 2

BF(0.34)

BF(0.34)

BF(0.34)

BF(0.34)

k = 3

BF(0.36)

BF(0.36)

BF(0.36)

BF(0.36)

Sum

BF(1.01)

BF(1.01)

BF(1.01)

BF(1.01)

Gc2

k = 1

BF(0.09)

BF(0.09)

WS(0.09)

WS(0.09)

k = 2

BF(0.09)

BF(0.09)

WS(0.09)

WS(0.09)

k = 3

BF(0.10)

BF(0.10)

WS(0.10)

WS(0.10)

Sum

BF(0.28)

BF(0.28)

WS(0.28)

WS(0.28)

Notes: ES = emission sources, where the abbreviations of emission source and candidate technology are consistent with those in Table 2 and Table 3, respectively. The number inside parentheses represents the treated amounts (t/d) of relevant technology.

Table 5. The optimal treated SO2 magnitude of various techniques

ES

T

w1 = 0.7 and w2 = 0.3

w1 = 0.6 and w2 = 0.4

w1 = 0.4 and w2 = 0.6

w1 = 0.3 and w2 = 0.7

PPc

k = 1

LG(69.31)

LG(69.31)

LG(69.31)

LG(69.31)

k = 2

LG(76.24)

LG(76.24)

LG(76.24)

LG(42.17)

CFB(34.07)

k = 3

LG(79.71)

LG(79.71)

LG(68.55)

CFB(11.16)

LG(68.69)

CFB(11.02)

Sum

LG(225.26)

LG(225.26)

LG(214.10)

CFB(11.16)

LG(180.17)

CFB(45.09)

PP

k = 1

CFB(47.70)

CFB(47.70)

CFB(47.70)

CFB(47.70)

k = 2

CFB(52.47)

CFB(52.47)

CFB(52.47)

CFB(52.47)

k = 3

CFB(54.86)

CFB(54.86)

CFB(54.86)

CFB(54.86)

Sum

CFB(155.03)

CFB(155.03)

CFB(155.03)

CFB(155.03)

Oc

k = 1

CFB(0.10)

CFB(0.10)

CFB(0.10)

CFB(0.10)

k = 2

CFB(0.11)

CFB(0.11)

CFB(0.11)

CFB(0.11)

k = 3

CFB(0.12)

CFB(0.12)

CFB(0.12)

CFB(0.12)

Sum

CFB(0.33)

CFB(0.33)

CFB(0.33)

CFB(0.33)

Gc1

k = 1

LG(1.62)

LG(1.62)

CFB(1.62)

CFB(1.62)

k = 2

LG(1.78)

CFB(1.78)

CFB(1.78)

CFB(1.78)

k = 3

LG(1.86)

CFB (1.86)

CFB (1.86)

CFB (1.86)

Sum

LG(5.26)

LG(1.62)

CFB(3.64)

CFB(5.26)

CFB(5.26)

Gc2

k = 1

CFB(0.08)

CFB(0.08)

CFB(0.08)

CFB(0.08)

k = 2

CFB(0.09)

CFB(0.09)

CFB(0.09)

CFB(0.09)

k = 3

CFB(0.09)

CFB(0.09)

CFB(0.09)

CFB(0.09)

Sum

CFB(0.26)

CFB(0.26)

CFB(0.26)

CFB(0.26)

Notes: ES = emission sources, where the abbreviations of emission source and candidate technology are consistent with those in Table 2 and Table 3, respectively. The number inside parentheses represents the treated amounts (t/d) of relevant technology.

Table 6. The optimal treated NOX magnitude of various techniques

ES

T

w1 = 0.7 and w2 = 0.3

w1 = 0.6 and w2 = 0.4

w1 = 0.4 and w2 = 0.6

w1 = 0.3 and w2 = 0.7

PPc

k = 1

LNB+SCR(14.54)

LNB+SCR(14.54)

LNB+SCR(14.54)

LNB+SCR(14.54)

k = 2

LNB+SCR(15.99)

LNB+SCR(15.99)

LNB+SCR(15.99)

LNB+SCR(15.99)

k = 3

LNB+SCR(16.72)

LNB+SCR(16.72)

LNB+SCR(16.72)

LNB+SCR(16.72)

Sum

LNB+SCR(47.25)

LNB+SCR(47.25)

LNB+SCR(47.25)

LNB+SCR(47.25)

PP

k = 1

LNB+SCR(70.69)

LNB+SCR(70.69)

LNB+SCR(70.69)

SCR(70.69)

k = 2

LNB+SCR(77.75)

SCR(77.75)

SCR(77.75)

SCR(77.75)

k = 3

LNB+SCR(81.29)

SCR(81.29)

SCR(81.29)

SCR(81.29)

Sum

LNB+SCR(229.73)

SCR(159.04) LNB+SCR(70.69)

SCR(159.04) LNB+SCR(70.69)

SCR(229.73)

Oc

k = 1

LNB+SCR(0.79)

LNB+SCR(0.79)

SCR(0.79)

SCR(0.79)

k = 2

SCR(0.86)

SCR(0.86)

SCR(0.86)

SCR(0.86)

k = 3

SCR(0.90)

SCR(0.90)

SCR(0.90)

SCR(0.90)

Sum

SCR(1.76)

LNB+SCR(0.79)

SCR(1.76)

LNB+SCR(0.79)

SCR(2.55)

SCR(2.55)

Gc1

k = 1

LNB+SCR(0.67)

LNB+SCR(0.67)

LNB+SCR(0.67)

LNB+SCR(0.67)

k = 2

LNB+SCR(0.74)

LNB+SCR(0.74)

LNB+SCR(0.74)

SCR(0.74)

k = 3

LNB+SCR(0.77)

LNB+SCR(0.77)

LNB+SCR(0.77)

SCR(0.77)

Sum

LNB+SCR(2.18)

LNB+SCR(2.18)

LNB+SCR(2.18)

SCR(1.51)

LNB+SCR(0.67)

Gc2

k = 1

LNB+SCR(0.63)

LNB+SCR(0.63)

LNB+SCR(0.63)

SCR(0.63)

k = 2

LNB+SCR(0.70)

SCR(0.70)

SCR(0.70)

SCR(0.70)

k = 3

LNB+SCR(0.73)

SCR(0.73)

SCR(0.73)

SCR(0.73)

Sum

LNB+SCR(2.06)

SCR(1.43)

LNB+SCR(0.63)

SCR(1.43)

LNB+SCR(0.63)

SCR(2.06)

Notes: ES = emission sources, where the abbreviations of emission source and candidate technology are consistent with those in Table 2 and Table 3, respectively. The number inside parentheses represents the treated amounts (t/d) of relevant technology.

COMMENT 13 -- (13) Page 17, Line 3, the unit t/a, is tons per?

RESPONSE: We regret for our unclear expression and have made the following revision (on Page 16):

“with treatment quantities of 3.84, 4.22, and 4.42 t/d”

COMMENT 14 -- (14) Page 17, line 43. Unbold “Figure 3”.

RESPONSE: We much appreciate the reviewer’s careful review and have made the correction as follows (on Page 17):

“Figure 4 reflects the residual PM2.5 concentration after the treatment process of four affected regions over three periods under nine weight combinations. ”

COMMENT 15 -- (15) You need to include the legend of Figure 3 y-axis.

RESPONSE: We much appreciate the reviewer’s careful review and have added the legend in the revised Figure 4 as follows (on Page 18):

Figure 4. The residual PM2.5 concentration of four affected regions over three periods.

COMMENT 16 -- (16) English spelling minor review.

RESPONSE: We accept the reviewer’s suggestion, and have polished the entire manuscript carefully.

Reviewer 2 Report

Review of An integration method for regional PM2.5 pollution control op- 2 timization based on meta-analysis and systematic review

I might not actually be the best person to review this paper, as the methods are outside of my fields of expertise. However, I think that the methods used are sound enough and the data sources diverse enough to warrant publication. My concerns regarding the paper center mostly around messaging. The writing is often meandering and ideas are presented in a manner that make reading the paper difficult. Instead of systematically building a case for their ideas, they seem to work backwards, expecting the reader to somehow infer the meanings by stating the conclusions of a paragraph before any other information has been presented.

I have very little comment on the science of the paper except to say that the authors do not properly concinvce the reader that there is a need for their ideas. They also do not indicate who is to benefit from their research. If their intended beneficiaries are in tourism related industries, then this should be explicitly stated.

The authors also do not present enough information on how these methods might have been applied successfully elsewhere. While I understand the multi outcome optimization models have been applied elsewhere, I do not know whether they have been applied to studies of health and economics as they appear to have been applied here. If not, have the methods been at least applied to some tangentially related topic?

Overall, this appears to be a reasonably good paper, but the authors should consider taking some time out to make it a bit more readable. If they do, then one would assume they would receive a wider readership.

Abstract:

Line 20: A meta analysis of what?

Line 23: multi objective optimization models usally balancing competing needs. This abstract never says what these competing needs are.

I find this abstract only minimally informative. The goals of the study are never clearly explained and have to be inferred from text later in the abstract. Require to follow a standard abstract format, clearly stat what the hypotheses of the study were and the data used. Summarize the results and then state your basic conclusions.

Introduction:

Line 42: Provide a reference to back this claim up.

Line 83: Provide a reference to back this up.

Table 1: Please clearly indicate what the parameter estimates are in the table legend. Also, what order are these presented in? Should they be arranged by region? Year? Region and year?

Line 193: I am assuming that the average daily PM concentrations are from the literature search. Explicitly state where these came from. The coefficient of exposure to reaction is from the literature search correct? This paragraph is somewhat difficult to read. I would suggest revising this paragraph to clearly present your methods. Stating your data source before the mathematical explanations might help.

Lines 209 – 216: Consider simplifying this text.

Line 366 There are two periods here.

Author Response

RESPONSES TO REVIEWERS’ COMMENTS

We are deeply grateful to the reviewers for their insightful reviews. The provided comments and suggestions have contributed much to improving the manuscript. According to them, we have made efforts to revise the manuscript, with the details explained as follows:

Note: For the reviewer’s convenience of re-review, we divided his/her comments into several parts with ordinal numbers, and corresponded to them in the following responses.

Responses to Reviewer #2’s comments

We are deeply grateful to the reviewers for their insightful reviews. The provided comments and suggestions have contributed much to improving the manuscript. According to them, we have made efforts to revise the manuscript, with the details explained as follows:

COMMENT 1 -- (1) I might not actually be the best person to review this paper, as the methods are outside of my fields of expertise. However, I think that the methods used are sound enough and the data sources diverse enough to warrant publication. My concerns regarding the paper center mostly around messaging. The writing is often meandering and ideas are presented in a manner that make reading the paper difficult. Instead of systematically building a case for their ideas, they seem to work backwards, expecting the reader to somehow infer the meanings by stating the conclusions of a paragraph before any other information has been presented.

RESPONSE: We appreciate the reviewer’s comments and have strengthened our presentation and writing by highlighting the main points at the end of essential paragraphs. Here are some examples (on Pages 2 to 6, 8 and 9, 13 and 14, 16 to 19):

“The severity of air pollution, especially the health threat of PM2.5 was emphasized in this paragraph.

The pollution degree of PM2.5 and its health damage in China were identified, which reflected the importance of health effect evaluation of this pollutant.

It clarified the importance of short-term treatment techniques and laid the foundation for the subsequent application of optimization model.

The successful application of the meta-analysis method in health effect evaluation demonstrated its feasibility and reliability. The health threat caused by PM2.5 pollution should be incorporated into the pollution control management.

The advantage of optimization model for pollution control and its potential improvement were pointed out in this paragraph.

Two important parts of this research work were described in this paragraph.

The main function of meta-analysis method was introduced.

This paragraph mainly illustrated how to complete the literature search.

The process of literature screening is described in this paragraph.

The software implementation of meta-analysis method and its output result are provided in this paragraph.

This section introduced the computational process of mortality caused by PM2.5 pollution, where the definition and source of each parameter were described in detail.

The section 3.2 described the formulation and solution process of two-objective optimization model. The definition of two objective functions, the implications of all constraints and the explanations of model parameters are provided in this section. The conversion of two-objective optimization model to single-objective one and the LINGO software are also introduced in this section.

The general introduction of Nanshan District and the necessity of regional air pollution control are provided in this paragraph.

From the perspective of system analysis, section 4.2 provides the detail information on system status, boundary, elements and critical parameters.

This paragraph introduced the nine weight combinations and emphasized the influence of weight combination on the treatment scheme.

It reflected the influence of weight combination on the utilization of PM-treated technology.

It reflected the influence of weight combination on the utilization of candidate treated technology of two pollutants (SO2 and NOx).

It described the variation in total treated quantities of various techniques in order to better reflect the influence of predetermined weight combinations on the treatment schemes.

The variation in residual PM2.5 concentration of affected regions under nine weight combinations and the related reasons are analyzed in this paragraph.

This paragraph described the variation in two objective functions and analyzed the trade-off between system economy and health risk.

The potential improvement of the proposed health effect evaluation model and optimization model for pollution control is discussed in this paragraph.”

COMMENT 2 -- (2) I have very little comment on the science of the paper except to say that the authors do not properly convince the reader that there is a need for their ideas. They also do not indicate who is to benefit from their research. If their intended beneficiaries are in tourism related industries, then this should be explicitly stated.

RESPONSE: We much appreciate the reviewer’s comments, and have clarified the raised issue as follows (on Recommendation):

“7. Recommendation

The serious economic loss, health threat and air quality degradation caused by PM2.5 pollution has aroused a wide attention globally. The research results of this study revealed the necessity of combing the health effect evaluation based on meta-analysis method and the air pollution control with aid of optimization model, and emphasized the importance of the balance between the high economic cost and people’s health threat. Some recommendations were provided based on the research findings: (i) for the local residents, it is necessary to obtain an in-depth insight into the health damage caused by PM2.5 pollution, which is beneficial to strengthen awareness of human health and environmental protection; (ii) for the administrators of government agency, this study helps them to identify the major pollutants sources and design effective pollution-mitigation measures. It is suggested that the health risk evaluation be incorporated into the process of air pollution control planning and management, with emission and ambient air quality standards being properly designed in order to avoid unnecessary socio-economic losses.”

COMMENT 3 -- (3) The authors also do not present enough information on how these methods might have been applied successfully elsewhere.

RESPONSE: We much appreciate the reviewer’s comments and regret for our unclear descriptions. As demonstrated in the section ‘literature review’, the successful applications of meta-analysis method and optimization model have proved their feasibility and applicability. According to the reviewer’s suggestions, we added the detailed description of the two methods, including their components, structure, formulation process and potential improvement, which laid a good theoretical foundation for their further advancement and applications in other fields. The effective application of this model in the Nanshan District of Shenzhen City, China, is expected to serve as a basis for similar work in other parts of the world in the future. Moreover, we have clarified the raised issue as follows (on Pages 2, 3,4 and 20):

“The existing research results have showed that the meta-analysis method can systematically evaluate and quantitatively analyze the health hazards caused by PM2.5 pollution by examining the existing results (Fang et al., 2016). A large number of epidemiological studies makes it possible to use meta-analysis to accurately calculate the exposure response coefficient, laying a foundation for quantifying the health effects of PM2.5 pollution (Atkinson et al., 2012; Bell et al., 2013; Atkinson et al., 2014; Atkinson et al., 2014; Chang et al., 2015; Cui et al., 2015; Lu et al., 2015; Luo et al., 2015; Tian et al., 2015; Zhao et al., 2015; Li et al., 2016; Sheehan et al., 2016; Song et al., 2016b; Achilleos et al., 2017; Chen et al., 2017; Li et al., 2017; Liu and Song, 2017; Nhung et al., 2017; Zhong et al., 2017; Kim et al., 2018; Liu et al., 2018; Liu et al., 2018; Requia et al., 2018; Vodonos et al., 2018; Fu et al., 2019; Karimi et al., 2019; Liang et al., 2020; Thayamballi et al., 2020).

This type of research often focuses on designing a suitable combination of pollution control measures by establishing the quantitative relationship between the emission of harmful substances and the atmosphere, and between cost and benefit (Carnevale et al., 2008; Pisoni and Volta, 2009; Carnevale et al., 2012; Zhen et al., 2016; Relvas et al., 2017; Sun et al., 2017; Yang and Teng, 2018; Wang and Yang, 2019; Xing et al., 2019; Huang et al., 2020). For example, Carnevale et al. (2008) established a nonlinear optimization model and applied it to the study of O3 pollution control in large cities in northern Italy, where model focused on the decontamination costs and improvement of the air quality index. Pisoni and Volta (2009) formulated a two-objective optimization model for tackling the pollution control issue of PM10, where the health impacts of PM10 pollution were considered as external cost.

  1. Method

In the ‘Methodology’ section, the section 3.1 introduced the formulation process of health evaluation model based on the meta-analysis method. Four critical steps, including the literature search, literature screening, determination of exposure-response relationship coefficient and the calculation of mortality caused by PM2.5 pollution, are described in detail. These are expected to set a good example for the application of meta-analysis method in other cases. The section 3.2 described the formulation and solution process of two-objective optimization model. The detailed introduction of model components, including two objective functions and all constraints, are provided in this section. Moreover, the conversion of two-objective optimization model to single-objective one and the LINGO software are also introduced to facilitate applications in other regions.

The effective application of this model in the Nanshan District of Shenzhen City, China, is expected to be a good example for similar work in other parts of the world in the future.”

COMMENT 4 -- (4) While I understand the multi outcome optimization models have been applied elsewhere, I do not know whether they have been applied to studies of health and economics as they appear to have been applied here. If not, have the methods been at least applied to some tangentially related topic?

RESPONSE: We much appreciate the reviewer’s comments. There are few studies that adopt multi-objective optimization model for concerning the health and economics of air pollution control (Pisoni and Volta, 2009; Carnevale et al., 2012; Relvas et al., 2017).

Pisoni, E., Volta, M., 2009. Modeling Pareto efficient PM10 control policies in Northern Italy to reduce health effects. ATMOSPHERIC ENVIRONMENT 43, 3243-3248.

Carnevale, C., Finzi, G., Pisoni, E., Volta, M., Wagber, F. 2012. Defining a nonlinear control problem to reduce particulate matter population exposure. ATMOSPHERIC ENVIRONMENT 55: 410-416.

Relvas, H., Miranda, A.I., Carnevale, C., Maffeis, G., Turrini, E., Volta, M. 2017. Optimal air quality policies and health: a multi-objective nonlinear approach. Environmental Science and Pollution Research 24, 13687-13699.

We also checked the references carefully and made all necessary corrections.

COMMENT 5 -- (5)Line 20: A meta analysis of what?

RESPONSE: We much appreciate the reviewer’s comments, and have clarified the raised issue as follows (on Abstract):

“Subjected to limited data information, it is assumed that the meta-analysis method through summarizing and combining the research results on the same subject was suitable to estimate the percentage of deaths caused by PM2.5 pollution.”

COMMENT 6 -- (6) Line 23: multi objective optimization models usually balancing competing needs. This abstract never says what these competing needs are.

RESPONSE: We much appreciate the reviewer’s comments, and have clarified the raised issue as follows (on Abstract):

“To balance the cost from air quality improvement and human health risks, a two-objective optimization model was developed.”

COMMENT 7 -- (7) I find this abstract only minimally informative. The goals of the study are never clearly explained and have to be inferred from text later in the abstract. Require to follow a standard abstract format, clearly stat what the hypotheses of the study were and the data used. Summarize the results and then state your basic conclusions.

RESPONSE: We accept the reviewer’s suggestion and have rewritten the Abstract as follows:

“ABSTRACT: PM2.5 pollution in China is becoming increasingly severe, threatening public health. The major goal of this study is to evaluate the mortality rate attributed to PM2.5 pollution and design pollution mitigation schemes in a southern district of China through a two-objective optimization model. The mortality rate is estimated by health effect evaluation model. Subjected to limited data information, it is assumed that the meta-analysis method through summarizing and combining the research results on the same subject was suitable to estimate the percentage of deaths caused by PM2.5 pollution. The critical parameters, such as the total number of deaths and the background concentration of PM2.5, were obtained through on-site survey, data collection, literature search, policy analysis and experts consultation. The equations for estimating the number of deaths caused by PM2.5 pollution was established by incorporating the relationship coefficient of exposure to reaction, calculated residual PM2.5 concentration of affected region and statistical total base number of deaths into a general framework. To balance the cost from air quality improvement and human health risks, a two-objective optimization model was developed. The first objective is to minimize the the mortality rate attributable to PM2.5 pollution and the second objective is to minimize the total system cost over three periods. The optimization results demonstrated that the combination of weights assigned to the two objectives significantly influenced the model output. For example, a high weight value assigned to minimizing the number of deaths results in the increased use of treatment techniques with higher efficiencies and a dramatic decrease in pollutant concentrations. In contrast, a model weighted more toward minimizing economic loss may lead to an increase in the death toll due to exposure to higher air pollution levels. The effective application of this model in the Nanshan District of Shenzhen City, China, is expected to serve as a basis for similar work in other parts of the world in the future.”

COMMENT 8 -- (8) Line 42: Provide a reference to back this claim up.

RESPONSE: We accept the reviewer’s suggestion and have cited the following related reference (on Page 1):

“Among many atmospheric pollutants, PM2.5 is believed to be the main culprit behind human morbidity/mortality (Xiao et al., 2021).”

“Xiao, Q.Y., Liang, F.C., Ning, M., Zhang, Q., Bi, J.Z., He, K.B., Lei, Y., Liu, Y. 2021. The long-term trend of PM2.5-related mortality in China: The effects of source data selection. CHEMOSPHERE 263, 127894.”

We also checked this reference carefully and made all necessary corrections.

COMMENT 9 -- (9) Line 83: Provide a reference to back this up.

RESPONSE: We accept the reviewer’s suggestion and have cited the following related reference (on Page 2):

“Existing research results show that the meta-analysis method can systematically evaluate and quantitatively analyze the health hazards caused by PM2.5 pollution (Fang et al., 2016).”

“Fang, D., Wang, Q.G., Li, H.M., Yu, Y.Y., Lu, Y., Qian, X. 2016. Mortality effects assessment of ambient PM2.5 pollution in the 74 leading cities of China. SCIENCE OF THE TOTAL ENVIRONMENT, 569, 1545-1552.”

We also checked the references carefully and made all necessary corrections.

COMMENT 10 -- (10) Table 1: Please clearly indicate what the parameter estimates are in the table legend.

RESPONSE: We much appreciate the reviewer’s comments, and have added the description of the parameter in the table legend as follows (on Page 5):

“Table 1. Information of the 15 peer-reviewed articles with the relationship coefficient of exposure to reaction (i.e. β) used for the meta-analysis”

COMMENT 11 -- (11) Also, what order are these presented in? Should they be arranged by region? Year? Region and year?

RESPONSE: We much appreciate the reviewer’s comments. The selected 15 articles are arranged in an alphabetical order of the first author. Accordingly, we have added the description of authors to Table 1 as follows (on Page 5):

Table 1. Information of 15 peer-reviewed articles with the relationship coefficient of exposure to reaction (i.e. β) used for the meta-analysis

Serial number of included literatures

Authors

Research area

Published period

β

95%CI

46

Song et al

Guangzhou

2012

0.009

(0.0055~ 0.0126)

47

Styszko et al

Shanghai

2013

0.0057

(0.0012~ 0.0101)

48

Sun et al

Shanghai

2013

0.0017

(0.0002~ 0.0035)

49

Thayamballi et al

Shanghai

2011

0.0047

(0.0022~ 0.0079)

50

Tian et al

Shanghai

2011

0.0047

(0.0022~ 0.0072)

51

Tsai et al

Shanghai

2013

0.0043

(0.0014~ 0.0073)

52

Vodonos et al

Guangzhou

2018

0.0055

(0.0024~ 0.0086)

53

Wang and Yang

Shenzhen

2016

0.0069

(0.0055~ 0.0083)

54

Wu et al

Fuzhou

2018

0.0017

(-0.0009~ 0.0043)

55

Wu et al

Changsha

2018

0.00518

(0.00065~ 0.00994)

56

Xie et al

Dongguan

2016

0.0052

(0.0024~ 0.008)

56

Xie et al

Foshan

2016

0.0091

(0.0061~ 0.0122)

56

Xie et al

Guangzhou

2016

0.0057

(0.0042~ 0.0073)

56

Xie et al

Jiangmen

2016

0.007

(0.0047~ 0.0093)

56

Xie et al

Shenzhen

2016

0.001

(-0.0004~ 0.0024)

56

Xie et al

Zhuhai

2016

0.0014

(-0.0006~ 0.0034)

57

Xing et al

Zhejiang Province

2018

0.0061

(0.0034~ 0.0089)

58

Yang et al

Pearl river delta

2017

0.0054

(0.0015~ 0.0092)

59

Yang et al

Huizhou

2017

0.0095

(0.0013~ 0.0179)

60

Yang and Teng

Guangzhou

2015

0.012

(0.0063~ 0.0177)

Notes: CI = Confidence interval.

COMMENT 12 -- (12) Line 193: I am assuming that the average daily PM concentrations are from the literature search. Explicitly state where these came from.

RESPONSE: We much appreciate the reviewer’s comments and regret for our unclear descriptions. Accordingly, we have added the following description to the revised manuscript (on Page 6):

 “C is the daily average PM2.5 concentration, which is calculated based on the conversion ratio between precursor pollutants (i.e. PM, SO2 and NOx) and PM2.5, and Gaussian dispersion model.”

Moreover, the daily average PM2.5 concentration equals to the left item of constraint (2i), which is written as follows:

(2i)

COMMENT 13 -- (13) The coefficient of exposure to reaction is from the literature search correct?

RESPONSE: Yes, it is from the literature search. The previous studies demonstrated that the determination of coefficient of exposure to reaction based on meta-analysis method is rational and feasible (Wang et al., 2012; Hoek et al., 2013; Fang et al., 2016; Fajersztajn et al., 2017; Xia et al., 2019). This is also the major advantage of meta-analysis method, while critical data information (i.e. the daily number of deaths caused by the PM2.5 pollution) is unavailable and is adverse to the identification of this coefficient. To ensure the robustness and rationality of estimated results provided by meta-analysis method, the comprehensive literature search and screening as well as the heterogeneity analysis of generated results based on statistical method are necessary.

Wang, D.Q., Wang, B.Q., Bai, Z.P. 2012. Meta-analysis of association between air fine particular and daily mortality of residents. Journal of Environment and Health, 29, 529-532.

Hoek, G., Krishnan, M.R., Beelen, R., Peters, A., Ostro, B., Brunekreef, B., Kaufman, J.D. 2013. Long term air pollution exposure and cardio-respiratory mortality: a review. ENVIRONMENTAL HEALTH, 12, 43.

Fang, D., Wang, Q.G., Li, H.M., Yu, Y.Y., Lu, Y., Qian, X. 2016. Mortality effects assessment of ambient PM2.5 pollution in the 74 leading cities of China. SCIENCE OF THE TOTAL ENVIRONMENT, 569, 1545-1552.

Fajersztajn, L., Saldiva, P., Pereira, L.A.A., Leite, V.F., Buehler, A.M. 2017. Short-term effects of fine particulate matter pollution on daily health events in Latin America: a systematic review and meta-analysis. INTERNATIONAL JOURNAL OF PUBLIC HEALTH, 62, 729-738.

Xia, Z., Wang, X.T., Yu, S.Y., Qian, Y. 2019. Meta-analysis of association between PM2.5 pollution and total non-accidental mortality of residents in different regions in China. Environmental Pollution & Control, 41, 891-895.

COMMENT 14 -- (14) This paragraph is somewhat difficult to read. I would suggest revising this paragraph to clearly present your methods. Stating your data source before the mathematical explanations might help.

RESPONSE: We agree with the reviewer’s comments and regret for our unclear descriptions. Accordingly, we have modified this paragraph in the revised manuscript as follows (on Page 5):

“Equations (1a) and (1b) were used to calculate the mortality rate of residents from exposure to PM2.5 pollution. The major parameters involved into this process were sourced from several ways. For example, the relationship coefficient of exposure to reaction (i.e. β) was generated by meta-analysis method; the average PM2.5 concentration of affected areas was calculated based on precursors’ conversion ratios to PM2.5 and Gaussian dispersion model; total base number of deaths for affected areas was collected from local statistical yearbook.”

COMMENT 15 -- (15) Lines 209–216: Consider simplifying this text.

RESPONSE: We much appreciate the reviewer’s comments, and have deleted the abundant contents as follows (on Page 6):

This study established a response relationship between pollution sources and affected areas to comprehensively consider the variations in regional air pollution, emission sources, economic and technical conditions of pollution regulations and control, as well as the requirements of pollution control. Following this, a A multi-objective optimization model aiming at minimizing the total pollution control cost and the number of deaths caused by PM2.5 pollution was developed under the constraints of environmental quality, and the economic and technical feasibility of control measures. The model is formulated as follows”

COMMENT 16 -- (16) Line 366 There are two periods here.

RESPONSE: We much appreciate the reviewer’s careful review and have corrected the indicated error as follows (on Page 12):

“The same was done using predetermined system boundaries, combined with the development goal and environmental problems of the Nanshan district.”

Generally, we deeply appreciate the reviewers and the editor for their insightful reviews. The provided comments/suggestions have contributed much to improving the manuscript.
